# Non-vacuous Bounds for the test error of Deep Learning without any change to the trained models

## Abstract

Deep neural network (NN) with millions or billions of parameters can perform really well on unseen data, after being trained from a finite training set. Various prior theories have been developed to explain such excellent ability of NNs, but do not provide a meaningful bound on the test error. Some recent theories, based on PAC-Bayes and mutual information, are non-vacuous and hence promising to explain the excellent performance of NNs. However, they often require a stringent assumption and extensive modification (e.g. compression, quantization) to the trained model of interest. Therefore, those prior theories provide a guarantee for the modified versions only. In this paper, we propose two novel bounds on the test error of a model. Our bounds uses the training set only and require no modification to the model. Those bounds are verified on a large class of modern NNs, pretrained by Pytorch on the ImageNet dataset, and are non-vacuous. To the best of our knowledge, these are the first non-vacuous bounds at this large scale, without any modification to the pretrained models.

## 1 Introduction

Deep neural networks (NNs) are arguably the most effective families in Machine Learning. They have been helping us to produce various breakthoughs, from mastering complex games [39], generating high-quality languages [10] or images [20], protein structure prediction [22], to building multi-task systems such as Gimini [41] and ChatGPT [1]. Big or huge NNs can efficiently learn knowledge from large datasets and then perform extremely well on unseen data.

Despite many empirical successes, there still remains a big gap between theory and practice of modern NNs. In particular, it is largely unclear [48] about *Why can deep NNs generalize well on unseen data after being trained from a finite number of samples?* This question relates to the generalization ability of a trained model. The standard learning theories suffer from various difficulties to provide a reasonable explanation. Various approaches have been studied, e.g. Radermacher complexity [18, 5], algorithmic stability [38, 11], algorithmic robustness [47, 40], PAC-Bayes [32, 7].

Some recent theories [50, 7, 28–30] are really promising, as they can provide meaningful bounds on the test error of some models. Dziugaite and Roy [14] obtained a non-vacuous bound by optimizing a distribution over NN parameters. [50, 16, 34] bounded the expected error of a *stochastic NN* by using off-the-shelf compression methods. Those theories follow the PAC-Bayes approach. On the other hand, Nadjahi et al. [35] showed the potential of the stability-based approach. Although making a significant progress, those theories are meaningful for small and *stochastic NNs* only.

Lotfi et al. [29, 30] made a significant step to analyze the generalization ability of big/huge NNs, such as large language models (LLM). Using state-of-the-art quantization, finetuning and some other

**Table 1:** Recent approaches for analyzing generalization error. ✓ means "Required" or "Yes". The upper part shows the required assumptions about different aspects, e.g., hypothesis space, loss function, training or finetuning. The lower part reports non-vacuousness in different situations.

| Approach | Radermacher complexity [5] | Alg. Stability [9, 27] | Alg. Robustness [47, 23, 42] | Mutual Info [46, 35] | PAC-Bayes [50, 34] | PAC-Bayes [29, 30] | Ours |
|---|---|---|---|---|---|---|---|
| **Requirement:** | | | | | | | |
| Model compressibility | | | | ✓ | ✓ | ✓ | |
| Train or finetune | | | | ✓ | ✓ | ✓ | |
| Lipschitz loss | ✓ | ✓ | | ✓ | | | |
| *Finite* hypothesis space | | | | | | | ✓ |
| **Non-vacuousness** for: | | | | | | | |
| *Stochastic* models only | | ✓ | | ✓ | ✓ | | |
| Trained models | | | | | | ✓ | ✓ |
| Training size > 1 M | | | | | | ✓ | ✓ |
| Model size > 500 M | | | | | | ✓ | ✓ |

techniques, the PAC-Bayes bounds by [30, 29] are non-vacuous for huge LLMs, e.g., GPT-2 and LLamMA2. Those bounds significantly push the frontier of deep learning theory.

In this work, we are interested in estimating or bounding the expected error $F(P, \boldsymbol{h})$ of a specific model (hypothesis) $\boldsymbol{h}$ which is trained from a finite number of samples from distribution $P$. The expected error tells how well a model $\boldsymbol{h}$ can generalize on unseen data, and hence can explain the performance of a trained model. This estimation problem is fundamental in learning theory [33], but arguably challenging for NNs. Many prior theories [50, 28, 35] were developed for *stochastic models*, but not for a trained model $\boldsymbol{h}$ of interest. Lotfi et al. [29, 30] made a significant progress to remove "stochasticity". For example, Lotfi et al. [30] provided a non-vacuous bound for the 2-bit quantized (and finetuned) versions of LLamMA2. Nonetheless, those theories require to use a method for intensively quantizing or compressing $\boldsymbol{h}$. This means that those theories are for the quantized or compressed models, and *hence may not necessarily be true for the original (unquantized or uncompressed) models*. This is a major limitation of those bounds. Such a limitation calls for novel theories that directly work with a given model $\boldsymbol{h}$.

Our contributions in this work are as follow:

- We develop a novel bound on the expected error $F(P, \boldsymbol{h})$ of a trained model $\boldsymbol{h}$. This bound does not require stringent assumptions as prior bounds do. It encodes both the complexity of the data distribution and the behavior of model $\boldsymbol{h}$ at local areas of the data space.
  The main technical challenge to obtain our bound is to use the training set to approximate an intractable term which summarizes the true error of $\boldsymbol{h}$ at different local areas of the data space. We resolve this challenge by analyzing various properties of small and binomial random variables.
- We next derive a tractable bound that can be easily computed from the training set only, without any change to $\boldsymbol{h}$. Hence this bound directly provides a guarantee for $\boldsymbol{h}$. Those properties are really beneficial and enable our bound to overcome the major limitations of prior theories. Table 1 presents a more detailed comparison about some key aspects.
- Third, we develop a novel bound that uses a data transformation method. This bound can help us to analyze more properties of a trained model, and enable an effective comparison between two trained models. This bound may be useful in many contexts, where prior theories cannot provide an effective answer.
- Finally, we did an extensive evaluation for a large class of modern NNs which were pretrained by Pytorch on the ImageNet dataset with more than 1.2M images. The results show that our bounds are non-vacuous. To the best of our knowledge, this is the first time that a theoretical bound is non-vacuous at this large scale, without any change to the trained models.

*Organization:* The next section presents a comprehensive survey about related work, the main advantages and limitations of prior theories. We then present our novel bounds in Section 3, accompanied with more detailed comparisons. Section 4 contains our empirical evaluation for some pretrained NNs. Section 5 concludes the paper. Proofs and more experimental details can be found in appendices.

*Notations:* $S$ often denotes a dataset and $|S|$ denotes its size/cardinality. $\Gamma$ denotes a partition of the data space. $[K]$ denotes the set $\{1, ..., K\}$ of natural numbers at most $K$. $\ell$ denotes a loss function, and $h$ often denotes a model or hypothesis of interest.

## 2 Related work

Various approaches have been studied to analyze generalization capability, e.g., Radermacher complexity [4], algorithmic stability [38, 15], algorithmic robustness [47], Mutual-infomation based bounds [46, 35], PAC-Bayes [32, 19]. Those approaches connect different aspects of a learning algorithm or hypothesis (model) to generalization.

**Norm-based bounds** [5, 18, 17] is one of the earliest approaches to understand NNs. The existing studies often use Rademacher complexity to provide data- and model-dependent bounds on the generalization error. An NN with smaller weight norms will have a smaller bound, suggesting better generalization on unseen data. Nonetheless, the norms of weight matrices are often large for practical NNs [3]. Therefore, most existing norm-based bounds are vacuous.

**Algorithmic stability** [9, 38, 12, 24] is a crucial approach to studying a learning algorithm. Basically, those theories suggest that a more stable algorithm can generalize better. Stable algorithms are less likely to overfit the training set, leading to more reliable predictions. The stability requirement in those theories is that a replacement of one sample for the training set will not significantly change the loss of the trained model. Such an assumption is really strong. One primary drawback is that achieving stability often requires restricting model complexity, potentially sacrificing predictive accuracy on challenging datasets. Therefore, this approach has a limited success in understanding deep NNs.

**Algorithmic robustness** [47, 40, 23, 42] is a framework to study generalization capability. It essesntially says that a robust learning algorithm can produce robust models which can generalize well on unseen data. This approach provides another lens to understand a learning algorithm and a trained model. However, it requires the assumption that the learning algorithm is robust, i.e., the loss of the trained model changes little in the small areas around the training samples. Such an assumption is really strong and cannot apply well for modern NNs, since many practical NNs suffer from adversarial attacks [31, 49]. Than et al. [42] showed that those theories are often vacuous.

**Neural Tangent Kernel** [21] provides a theoretical lens to study generalization of NNs by linking them to kernel methods in the infinite-width limit. As networks grow wider, their training dynamics under gradient descent can be approximated by a kernel function which remains constant throughout training. This perspective simplifies the analysis of complex neural architectures. The framework enables explicit generalization bounds, and a deeper understanding of how network architecture and initialization affect learning. However, the main limitation of this framework comes from its assumptions, such as the *infinite-width* regime and fixed kernel during training, may not fully capture the behavior of finite, practical networks where feature learning is dynamic. Some other studies [25] can remove the infinite-width regime but assume the *infinite depth*.

**Mutual information (MI)** [46, 35] has emerged as a powerful tool for analyzing generalization by quantifying the dependency between a model's learned representations and the data. Since a trained model contains the (compressed) knowledge learned from the training samples, MI offers a principled framework for studying the trade-off between compression and predictive accuracy. However, the existing MI-based theories [46, 45, 37, 35] have a notable drawback: computing MI in high-dimensional, non-linear settings is computationally challenging. This drawback poses significant challenges for analyzing deep NNs, although [35] obtained some promising results on small NNs.

**PAC-Bayes** [32, 19, 8] recently has received a great attention, and provide non-vacuous bounds [50, 34] for some NNs. Those bounds often estimate $\mathbb{E}_{\hat{h}}[F(P, \hat{h})]$ which is the expectation of the test error over the posterior distribution of $\hat{h}$. It means that those bounds are for a *stochastic model* $\hat{h}$. Hence they provide limited understanding for a specific deterministic model $h$. Neyshabur et al. [36] provided an attempt to derandomization for PAC-Bayes but resulted in vacuous bounds for modern neural networks [3]. Some recent attempts to derandomization include [44, 13].

**Non-vacuous bounds for NNs:** Dziugaite and Roy [14] obtained a non-vacuous bound for NNs by finding a posterior distribution over neural network parameters that minimizes the PAC-Bayes bound.

Their optimized bound is non-vacuous for a stochastic MLP with 3 layers trained on MNIST dataset. Zhou et al. [50] bounded the population loss of a stochastic NNs by using compressibility level of a NN. Using off-the-shelf neural network compression schemes, they provided the first non-vacuous bound for LeNet-5 and MobileNet, trained on ImageNet with more than 1.2M samples. Lotfi et al. [28] developed a compression method to further optimize the PAC-Bayes bound, and estimated the error rate of 40.9% for MobileViT on ImageNet. Mustafa et al. [34] provided a non-vacuous PAC-Bayes bound for adversarial population loss for VGG on CIFAR10 dataset. Galanti et al. [16] presented a PAC-Bayes bound which is non-vacuous for Convolutional NNs with up to 20 layers and for CIFAR10 and MNIST. Akinwande et al. [2] provided a non-vacuous PAC-Bayes bound for prompts. Although making a significant progress for NNs, those bounds are non-vacuous for stochastic neural networks only. Biggs and Guedj [7] provided PAC-Bayes bounds for deterministic models and obtain (empirically) non-vacuous bounds for a specific class of (SHEL) NNs with a single hidden layer, trained on MNIST and Fashion-MNIST. Nonetheless, it is unclear about how well those bounds apply to bigger or deeper NNs.

Towards understanding big/huge NNs, Lotfi et al. [29, 30] made a significant step that provides non-vacuous bounds for LLMs. While the PAC-Bayes bound in [29] can work with LLMs trained from i.i.d data, the recent bound in [30] considers token-level loss for LLMs and applies to dependent settings, which is close to the practice of training LLMs. Using both model quantization, finetuning and some other techniques, the PAC-Bayes bound by [30] is shown to be non-vacuous for huge LLMs, e.g., LLamMA2. Those bounds significantly push the frontier of learning theory towards building a solid foundation for DL.

Nonetheless, there are two main drawbacks of those bounds [29, 30]. First, model quantization or compression is required in order to obtain a good bound. It means, those bounds are for the quantized or compressed models, and *hence may not necessarily be true for the original (unquantized or uncompressed) models*. For example, [30] provided a non-vacuous bound for the 2-bit quantized versions of LLamMA2, instead of their original pretrained versions. Second, those bounds require the assumption that *the model (hypothesis) family is finite*, meaning that a learning algorithm only searches in a space with finite number of specific models. Although such an assumption is reasonable for the current computer architectures, those bounds cannot explain a trained model that belongs to families with infinite (or uncountable) number of members, which are provably prevalent. In contrast, our bounds apply directly to any specific model without requiring any modification or support. A comparison between our bounds and prior approaches about some key aspects is presented in Table 1.

## 3 Error bounds

In this section, we present three novel bounds for the expected error of a given model. The first bound provides a general form which directly depends on the complexity of the data distribution and the trained model. The second bound provides an explicit upper bound for the error, which can be computed directly from any given dataset. The last bound helps us to analyze the robustness of a model by using data augmentation.

Consider a hypothesis (or model) $\boldsymbol{h}$, defined on an instance set $\mathcal{Z}$, and a nonnegative loss function $\ell$. Each $\ell(\boldsymbol{h}, \boldsymbol{z})$ tells the loss (or quality) of $\boldsymbol{h}$ at an instance $\boldsymbol{z} \in \mathcal{Z}$. Given a distribution $P$ defined on $\mathcal{Z}$, the quality of $\boldsymbol{h}$ is measured by its *expected loss* $F(P, \boldsymbol{h}) = \mathbb{E}_{\boldsymbol{z} \sim P}[\ell(\boldsymbol{h}, \boldsymbol{z})]$. Quantity $F(P, \boldsymbol{h})$ tells the generalization ability of model $\boldsymbol{h}$; a smaller $F(P, \boldsymbol{h})$ implies that $\boldsymbol{h}$ can generalize better on unseen data.

For analyzing generalization ability, we are often interested in estimating (or bounding) $F(P, \boldsymbol{h})$. Sometimes this expected loss is compared with the *empirical loss* of $\boldsymbol{h}$ on a data set $\boldsymbol{S} = \{\boldsymbol{z}_1, ..., \boldsymbol{z}_n\} \subseteq \mathcal{Z}$, which is defined as $F(\boldsymbol{S}, \boldsymbol{h}) = \frac{1}{n} \sum_{\boldsymbol{z} \in \boldsymbol{S}} \ell(\boldsymbol{h}, \boldsymbol{z})$. Note that a small $F(\boldsymbol{S}, \boldsymbol{h})$ does not neccessarily imply good generalization of $\boldsymbol{h}$, since overfitting may appear. Therefore, our ultimate goal is to estimate $F(P, \boldsymbol{h})$ directly.

Let $\Gamma(\mathcal{Z}) := \bigcup_{i=1}^{K} \mathcal{Z}_i$ be a partition of $\mathcal{Z}$ into $K$ disjoint nonempty subsets. Denote $\boldsymbol{S}_i = \boldsymbol{S} \cap \mathcal{Z}_i$, and $n_i = |\boldsymbol{S}_i|$ as the number of samples falling into $\mathcal{Z}_i$, meaning that $n = \sum_{j=1}^{K} n_j$. Denote $\boldsymbol{T} = \{i \in [K] : n_i > 0\}, a_i(\boldsymbol{h}) = \mathbb{E}_{\boldsymbol{z}}[\ell(\boldsymbol{h}, \boldsymbol{z}) | \boldsymbol{z} \in \mathcal{Z}_i]$ for $i \in [K]$, and $a_o = \max_{j \notin \boldsymbol{T}} a_j(\boldsymbol{h})$.

## 3.1 General bound

The first result incorporates the properties of the data distribution and the trained model.

**Theorem 3.1.** *Given a partition $\Gamma$ and a bounded nonnegative loss $\ell$, consider a model $\boldsymbol{h}$ which may depend on a dataset $\boldsymbol{S}$ with $n$ i.i.d. samples from distribution $P$. Denote $p_i = P(\mathcal{Z}_i)$ as the measure of area $\mathcal{Z}_i$ for $i \in [K]$, and $u = \sum_{i=1}^{K} \gamma n p_i(1 + \gamma n p_i)$. For any constants $\gamma \geq 1$, $\delta_1 \geq \exp(-\frac{u \ln \gamma}{4n-3})$ and $\delta_2 > 0$, we have the following with probability at least $1 - \delta_1 - \delta_2$:*

$$F(P, \boldsymbol{h}) \leq F(\boldsymbol{S}, \boldsymbol{h}) + C\sqrt{\frac{u}{2n^2} \ln \frac{1}{\delta_1}} + g(\Gamma, \boldsymbol{h}, \delta_2) \tag{1}$$

*where $g(\Gamma, \boldsymbol{h}, \delta_2) = \frac{\sqrt{\ln(2K/\delta_2)}}{n} \sum_{i \in \boldsymbol{T}} \sqrt{n_i} \left(a_o + \sqrt{2}a_i(\boldsymbol{h})\right) + \frac{2\ln(2K/\delta_2)}{n}(a_o|\boldsymbol{T}| + \sum_{i \in \boldsymbol{T}} a_i(\boldsymbol{h}))$ and $C = \sup_{\boldsymbol{z} \in \mathcal{Z}} \ell(\boldsymbol{h}, \boldsymbol{z})$.*

This theorem suggests that the expected loss cannot be far from the empirical loss $F(\boldsymbol{S}, \boldsymbol{h})$. The gap between the two is at most $C\sqrt{\frac{u}{2n^2} \ln \frac{1}{\delta_1}} + g(\Gamma, \boldsymbol{h}, \delta_2)$. Such a gap represents the uncertainty of our bound and mostly depends on the sample size $n$, the trained model $\boldsymbol{h}$, the data distribution $P$ and the partition $\Gamma$. We emphasize that bound (1) has some interesting properties:

- *First, it does not require any assumption about the hypothesis family and learning algorithm.* This is an advantage over many approaches including algorithmic stability [9, 27], robustness [47, 23], Radermacher complexity [4, 5]. This bound focuses directly on the the model $\boldsymbol{h}$ of interest, helping it to be tighter than many prior bounds.
- *Second, it depends on the complexity of the data distribution.* Note that $u$ encodes the complexity of $P$. For a uniform partition $\Gamma$, a more structured distribution $P$ can have a higher sum $\sum_{i=1}^{K} p_i^2$. As an example of structured distributions, a Gaussian with a small variance has the most probability density in a small area around its mean and lead to a high $p_i$ for some $i$. Meanwhile a less structured distribution (e.g. uniform) can produce a small $\sum_{i=1}^{K} p_i^2$ and hence smaller $u$. To the best of our knowledge, such an explicit dependence on the distribution complexity is rare in prior theories.
- *Third, it is model-dependent.* Some particular properties of model $\boldsymbol{h}$ are encoded in $g(\Gamma, \boldsymbol{h}, \delta_2)$ and the empirical loss . A better model $\boldsymbol{h}$ will lead to smaller $a_i$'s and hence $g$. On the other hand, a worse model can have a bigger $g$, leading to a higher RHS of (1).

It is worth noticing the similarity between our bound (1) and robustness-based bounds in [23, 42]. $F(\boldsymbol{S}, \boldsymbol{h}) + g(\Gamma, \boldsymbol{h}, \delta_2)$ is the common part in those bounds. Our bound (1) contains $C\sqrt{\frac{u}{2n^2} \ln \frac{1}{\delta_1}}$ that encodes the complexity of the data distribution, whereas the bounds in [23, 42] use a robustness quantity that measures the sensitivity of the loss w.r.t. a change in the input. While prior bounds are not amenable to be exactly computed from a training set, our bound enables to easily derive a computable and non-vacuous bound (below). This is the main advantage of bound (1).

*Proof sketch.* The detailed proof can be found in Appendix A. We focus on bounding the probability $\Pr(F(P, \boldsymbol{h}) - F(\boldsymbol{S}, \boldsymbol{h}) \geq \phi)$, for some gap $\phi$. Note that $F(P, \boldsymbol{h}) - F(\boldsymbol{S}, \boldsymbol{h}) = A + B$, where $A = F(P, \boldsymbol{h}) - \sum_i \frac{n_i}{n} a_i(\boldsymbol{h})$ and $B = \sum_i \frac{n_i}{n} a_i(\boldsymbol{h}) - F(\boldsymbol{S}, \boldsymbol{h})$. Therefore, our proof estimates $\Pr(A \geq g)$ and

$$\Pr(B \geq t) \tag{2}$$

for some constant $t$. Once they are known, we can use the union bound to obtain a bound on $\Pr(F(P, \boldsymbol{h}) - F(\boldsymbol{S}, \boldsymbol{h}) \geq g + t)$ as desired. We use a result from [23] to bound $\Pr(A \geq g)$. The remaining task is to estimate (2), which is **the main challenge**. This challenge requires approximating an intractable quantity from a data set.

We resolve this challenge by developing Theorem A.1. Its proof contains three main steps:

1. First we show $\Pr(B \geq t) \leq e^{-yt}\mathbb{E}_{\boldsymbol{h},\boldsymbol{n}}\left[\mathbb{E}_{\boldsymbol{S}}\left[e^{yB}|\boldsymbol{h},\boldsymbol{n}\right]\right]$, for $\boldsymbol{n} = \{n_1, ..., n_K\}$ and some $y$.

2. We next estimate $\mathbb{E}_{\boldsymbol{S}}\left[e^{yB_K}|\boldsymbol{h},\boldsymbol{n}\right]$. Overall, we make sure that $\mathbb{E}_{\boldsymbol{S}}\left[e^{yB}|\boldsymbol{h},\boldsymbol{n}\right] \leq e^{\psi(y,\boldsymbol{n})}$, for some function $\psi(y, \boldsymbol{v})$ which does not depend on $\boldsymbol{h}$. As a result $\Pr(B \geq t) \leq \mathbb{E}_{\boldsymbol{v}}e^{\psi(y,\boldsymbol{n})}$.

3. The last step is to bound $\mathbb{E}_{\boldsymbol{n}}e^{\psi(y,\boldsymbol{n})}$. This requires us to develop various analyses for small random variables in Appendix B. A suitable choice for $t, y$ completes our proof. $\qquad \square$

## 3.2 Tractable bounds

It is worth noticing that bound (1) contains some unknown quantities, e.g., $u$ and $a_i$'s, which cannot be computed exactly. This is its main limitation. The following bound overcomes such a limitation.

**Theorem 3.2.** *Given the notations and assumption in Theorem 3.1, for any constants $\gamma \geq 1, \delta > 0$ and $\alpha \in [0, \frac{\gamma n(K + \gamma n)}{K(4n-3)}]$, we have the following with probability at least $1 - \gamma^{-\alpha} - \delta$:*

$$F(P, \boldsymbol{h}) \leq F(\boldsymbol{S}, \boldsymbol{h}) + C\sqrt{\hat{u}\alpha \ln \gamma} + g_2(\delta/2) \tag{3}$$

*where $\hat{u} = \frac{\gamma}{2n} + \frac{\gamma^2}{2} \sum_{i=1}^{K} \left(\frac{n_i}{n}\right)^2 + \gamma^2 \sqrt{\frac{2}{n} \ln \frac{2K}{\delta}}$, $g_2(\delta) = \frac{C(1+\sqrt{2})\sqrt{\ln(2K/\delta)}}{n} \sum_{i \in \boldsymbol{T}} \sqrt{n_i} + \frac{4C|\boldsymbol{T}|\ln(2K/\delta)}{n}$.*

One special property is that *we can evaluate our bound easily by using only the training set*. Indeed, we can choose $K$ and a specific partition $\Gamma$ of the data space. Then we can count $n_i$ and $\boldsymbol{T}$ and evaluate the bound (3) easily. This property is remarkable and beneficial in practice.

**A theoretical comparison with closely related bounds:** Although many model-dependent bounds [23, 42, 7, 44, 29, 30] have been proposed, our bound (3) has various advantages:

- *Mild assumption:* Our bound does not require stringent assumptions as in prior ones. Some prior bounds require stability [27, 26] or robustness [47, 23, 40] of the learning algorithm. Those assumptions are often violated in practice, e.g. for the appearance of adversarial attacks [49]. Some theories [29, 30] assume that the hypothesis class is finite, which is restrictive. In contrast, our bound requires only i.i.d. assumption which also appears in most prior bounds.
- *Easy evaluation:* An evaluation of our bound (3) will be simple and does not require any modification to the model $\boldsymbol{h}$ of interest. This is a crucial advantage. Many prior theories require intermediate steps to change the model of interest into a suitable form. For example, state-of-the-art methods to compress NNs are required for [50, 28, 35]; quantization for a model is required for [29, 30]; finetuning (e.g. SubLoRA) is required for [29, 30]. Those facts suggests that evaluations for prior bounds are often expensive. Besides, many prior model-dependent bounds [47, 23, 42] cannot be exactly computed from a training set only.
- *No change to the model:* Most prior non-vacuous bounds [50, 14, 29, 30] require extensively compressing (or quantizing) model $\boldsymbol{h}$ of interest and then retraining/finetuning the compressed version. Sometimes the compression step is too restrictive and produces low-quality models [29]. Therefore, a modification will change model $\boldsymbol{h}$ and hence ***those bounds do not directly provide guarantees for the generalization ability of $\boldsymbol{h}$***. In contrast, our bound (3) does not require any change to model $\boldsymbol{h}$, and hence directly provides a guarantee for $\boldsymbol{h}$.

There is a nonlinear relationship between $K$ and the uncertainty term $\mathrm{Unc}(\Gamma) = C\sqrt{\hat{u}\alpha \ln \gamma} + g_2(\delta/2)$ in our bound. A partition with a larger $K$ can make the sum $\sum_{i=1}^{K} \left(\frac{n_i}{n}\right)^2$ smaller, as the samples can be spread into more areas. However a larger $K$ can make $g_2(\delta)$ larger. Therefore, we should not choose too large $K$. On the other hand, a small $K$ can make the sum $\sum_{i=1}^{K} \left(\frac{n_i}{n}\right)^2$ large, since more samples can appear in each area $\mathcal{Z}_i$ and enlarge $\frac{n_i}{n}$. Therefore, we should not choose too small $K$. Furthermore, we need to choose constant $\alpha$ carefully, since there is a trade-off in the bound and the certainty $1 - \gamma^{-\alpha} - \delta$. A smaller $\alpha$ can make the bound smaller, but could enlarge $\gamma^{-\alpha}$ and hence reduce the certainty of the bound.

The next result considers the robustness of a model.

**Theorem 3.3.** *Given the assumption in Theorem 3.2, let $\hat{\boldsymbol{S}} = \mathcal{T}(\boldsymbol{S})$ be the result of using a transformation method $\mathcal{T}$, which is independent with $\boldsymbol{h}$, on the samples of $\boldsymbol{S}$. Denote $\bar{\epsilon}(\boldsymbol{h}) = \sum_{i \in \boldsymbol{T}} \frac{m_i}{m} \bar{\epsilon}_i$ and $m = \sum_{i \in \boldsymbol{T}} m_j$, where $\hat{\boldsymbol{S}}_i = \hat{\boldsymbol{S}} \cap \mathcal{Z}_i$, $m_i = |\hat{\boldsymbol{S}}_i|$, and $\bar{\epsilon}_i = \frac{1}{m_i n_i} \sum_{\boldsymbol{z} \in \boldsymbol{S}_i, \boldsymbol{s} \in \hat{\boldsymbol{S}}_i} |\ell(\boldsymbol{h}, \boldsymbol{z}) - \ell(\boldsymbol{h}, \boldsymbol{s})|$ for each $i \in \boldsymbol{T}$. We have the following with probability at least $1 - \gamma^{-\alpha} - \delta$:*

$$F(P, \boldsymbol{h}) \leq \bar{\epsilon}(\boldsymbol{h}) + F(\hat{\boldsymbol{S}}, \boldsymbol{h}) + \sum_{i \in \boldsymbol{T}} \left(\frac{n_i}{n} - \frac{m_i}{m}\right) F(\boldsymbol{S}_i, \boldsymbol{h}) + C\sqrt{\hat{u}\alpha \ln \gamma} + g_2(\delta/2) \tag{4}$$

This theorem suggests that *a model can be better if its loss is less sensitive with respect to some small changes in the training samples*. This can be seen from each quantity $\bar{\epsilon}_i$ which measures the average

267 difference of the loss of $\boldsymbol{h}$ for the samples $\boldsymbol{S}_i$ and $\hat{\boldsymbol{S}}_i$ belonging to the same small area. This result
268 closely relates to adversarial training [31], where one often wants to train a model which is robust
269 w.r.t small changes in the inputs. It is also worth noticing that if $\mathcal{T}$ transforms $\boldsymbol{S}$ too much, both the
270 loss $F(\hat{\boldsymbol{S}}, \boldsymbol{h})$ and the sensitivity $\bar{\epsilon}$ can be large. As a result, the bound (4) will be large. In fact, our
271 proof suggests that bound (4) is worse than bound (3).

272 The main benefit of Theorem 3.3 is that we can use some transformation methods to compare some
273 trained models. This is particularly useful for the cases where two models have comparable (even
274 zero) training losses. For those cases, Theorem 3.2 does not provide a satisfactory answer. Instead, we
275 can use a simple augmentation method (e.g., noise perturbation, rotation, translation, ...) to produce
276 a dataset $\hat{\boldsymbol{S}}$ and then use this dataset to evaluate the upper bound (4). By this way, we use both the
277 training loss $F(\boldsymbol{S}, \boldsymbol{h})$ and $\bar{\epsilon}(\boldsymbol{h}) + F(\hat{\boldsymbol{S}}, \boldsymbol{h}) + \sum_{i \in \boldsymbol{T}} \left( \frac{n_i}{n} - \frac{m_i}{m} \right) F(\boldsymbol{S}_i, \boldsymbol{h})$ for comparison.

# 4   Empirical evaluation

279 In this section, we present two sets of extensice evaluations about the our bounds. We use 32 modern
280 NN models[1] which were pretrained by Pytorch on the ImageNet dataset with 1,281,167 images.
281 Those models are multiclass classifiers. Our main aim is to provide a guarantee for the error of a
282 trained model, without any further modification. Therefore, no prior bound is taken into comparison,
283 since those existing bounds are either already vacuous or require some extensive modifications or
284 cannot directly apply to those trained NNs.

## 4.1   Large-scale evaluation for pretrained models

286 The first set of experiments verifies nonvacuouness of our first bound (3) and the effects of some
287 parameters in the bound. We use the training part of ImageNet only to compute the bound.

288 **Experimental settings:** We fix $\delta = 0.01, \alpha = 100, \gamma = 0.04^{-1/\alpha}$. This choice means that our
289 bound is correct with probability at least 95%. The partition $\Gamma$ is chosen with $K = 200$ small areas
290 of the input space, by clustering the training images into 200 areas, whose centroids are initialized
291 randomly. The upper bound (3) for each model was computed with 5 random seeds. We use the 0-1
292 loss function, meaning that our bound directly estimates the true classification error.

293 **Results:** The overall results are reported in Table 2. One can observe that our bound for all models
294 are all non-vacuous even for the non-optimized choices of some parameters. Our estimate is often
295 2-3 times higher than the oracle test error of each model. When choosing the best parameter for
296 each model by grid search, we can obtain much better bounds about the test errors. Note that
297 non-vacuousness of our bound holds true for a large class of deep NN families, some of which have
298 more than 630M parameters. To the best of our knowledge, bound (3) is the first theoretical bound
299 which is non-vacuous at such a large scale, without requiring any modification to the trained models.

300 **Effect of parameters:** Note that our bound depends on the choice of some parameters. Figure 1
301 reports the changes of $\sum_{i=1}^{K} \left( \frac{n_i}{n} \right)^2$ as the partition $\Gamma$ changes. We can see that this quantity tends
302 to decrease as we divide the input space into more small areas. Meanwhile, Figure 2 reports the
303 uncertainty term, as either $\alpha$ or $K$ changes. Observe that a larger $K$ can increase the uncertainty fast,
304 while an increase in $\alpha$ can gradually decrease the uncertainty. Those figures enable an easy choice
305 for the parameters in our bound.

## 4.2   Evaluation with data augmentation

307 As mentioned before, our bound (3) can provide a theoretical certificate for a trained model, but may
308 not be ideal to compare two models which have the same training error. Sometimes, a model can
309 have a lower training error but a higher test error (such as DenseNet161 vs. DenseNet201, VIT L 16
310 linear vs. VIT L 16 V1). Bound (3) may not be good for model comparison. In those cases, we need
311 to use bound (4) for comparison.

312 **Experimental settings:** We fix $\delta = 0.01, \alpha = 100, \gamma = 0.04^{-1/\alpha}, K = 200$ as before. We use
313 white noise addition as the transformation method in Theorem 3.3. Specifically, each image is added

---

[1]https://pytorch.org/vision/stable/models.html

**Table 2:** Upper bounds on the true error (in %) of 32 deep NNs which were pretrained on ImageNet dataset. The second column presents the model size, the third column contains the test accuracy at Top 1, as reported by Pytorch. "Mild" reports the bound for the choice of $\{\delta = 0.01, K = 200, \alpha = 100, \gamma = 0.04^{-1/\alpha}\}$, while "Optimized" reports the bound with parameter optimization by grid search. The grid search is done for $K \in \{100, 200, 300, 400, 500, 1000, 5000, 10000\}, \alpha \in \{10, 20, ..., 100\}, \delta = 0.01$ and $\gamma = 0.04^{-1/\alpha}$. The last two columns report our estimates about the true error, with a certainty at least 95%.

| Model | #Params (M) | Training error | Acc@1 | Test error | Error bound (3) Mild | Optimized |
|---|---|---|---|---|---|---|
| ResNet18 V1 | 11.7 | 21.245 | 69.758 | 30.242 | 57.896 ±4.189 | 54.262 |
| ResNet34 V1 | 21.8 | 15.669 | 73.314 | 26.686 | 52.320 ±4.189 | 48.686 |
| ResNet50 V1 | 25.6 | 13.121 | 76.130 | 23.870 | 49.772 ±4.189 | 46.138 |
| ResNet101 V1 | 44.5 | 10.502 | 77.374 | 22.626 | 47.153 ±4.189 | 43.519 |
| ResNet152 V1 | 60.2 | 10.133 | 78.312 | 21.688 | 46.784 ±4.189 | 43.150 |
| ResNet50 V2 | 25.6 | 8.936 | 80.858 | 19.142 | 45.587 ±4.189 | 41.953 |
| ResNet101 V2 | 44.5 | 6.008 | 81.886 | 18.114 | 42.659 ±4.189 | 39.025 |
| ResNet152 V2 | 60.2 | 5.178 | 82.284 | 17.716 | 41.829 ±4.189 | 38.195 |
| SwinTransformer B | 87.8 | 6.464 | 83.582 | 16.418 | 43.115 ±4.189 | 39.481 |
| SwinTransformer B V2 | 87.9 | 6.392 | 84.112 | 15.888 | 43.043 ±4.189 | 39.409 |
| SwinTransformer T | 28.3 | 9.992 | 81.474 | 18.526 | 46.643 ±4.189 | 43.009 |
| SwinTransformer T V2 | 28.4 | 8.724 | 82.072 | 17.928 | 45.375 ±4.189 | 41.741 |
| VGG13 | 133.0 | 18.456 | 69.928 | 30.072 | 55.107 ±4.189 | 51.473 |
| VGG13 BN | 133.1 | 19.223 | 71.586 | 28.414 | 55.874 ±4.189 | 52.240 |
| VGG19 | 143.7 | 16.121 | 72.376 | 27.624 | 52.772 ±4.189 | 49.138 |
| VGG19 BN | 143.7 | 15.941 | 74.218 | 25.782 | 52.592 ±4.189 | 48.958 |
| DenseNet121 | 8.0 | 15.631 | 74.434 | 25.566 | 52.282 ±4.189 | 48.648 |
| DenseNet161 | 28.7 | 10.48 | 77.138 | 22.862 | 47.131 ±4.189 | 43.497 |
| DenseNet169 | 14.1 | 12.395 | 75.600 | 24.400 | 49.046 ±4.189 | 45.412 |
| DenseNet201 | 20.0 | 9.806 | 76.896 | 23.104 | 46.457 ±4.189 | 42.823 |
| ConvNext Base | 88.6 | 5.209 | 84.062 | 15.938 | 41.860 ±4.189 | 38.226 |
| ConvNext Large | 197.8 | 3.846 | 84.414 | 15.586 | 40.497 ±4.189 | 36.863 |
| RegNet Y 128GF e2e | 644.8 | 5.565 | 88.228 | 11.772 | 42.216 ±4.189 | 38.582 |
| RegNet Y 128GF linear | 644.8 | 9.032 | 86.068 | 13.932 | 45.683 ±4.189 | 42.049 |
| RegNet Y 32GF e2e | 145.0 | 7.127 | 86.838 | 13.162 | 43.778 ±4.189 | 40.144 |
| RegNet Y 32GF linear | 145.0 | 10.558 | 84.622 | 15.378 | 47.209 ±4.189 | 43.575 |
| RegNet Y 32GF V2 | 145.0 | 3.761 | 81.982 | 18.018 | 40.412 ±4.189 | 36.778 |
| VIT B 16 linear | 86.6 | 14.969 | 81.886 | 18.114 | 51.620 ±4.189 | 47.986 |
| VIT B 16 V1 | 86.6 | 5.916 | 81.072 | 18.928 | 42.567 ±4.189 | 38.933 |
| VIT H 14 linear | 632.0 | 9.951 | 85.708 | 14.292 | 46.602 ±4.189 | 42.968 |
| VIT L 16 linear | 304.3 | 11.003 | 85.146 | 14.854 | 47.654 ±4.189 | 44.020 |
| VIT L 16 V1 | 304.3 | 3.465 | 79.662 | 20.338 | 40.116 ±4.189 | 36.482 |

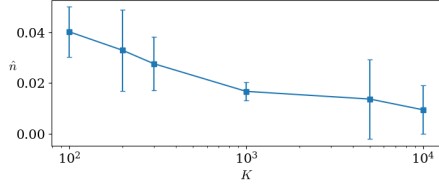

**Figure 1:** The dynamic of $\hat{n} = \sum_{i=1}^{K} \left(\frac{n_i}{n}\right)^2$ as $K$ changes.

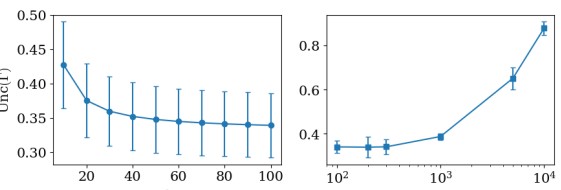

**Figure 2:** The uncertainty $\text{Unc}(\Gamma) = C\sqrt{\hat{u}\alpha \ln \gamma} + g(\delta/2)$ as (right) $K$ changes and (left) $\alpha$ changes, for fixed $K = 200, \gamma = 0.04^{-1/\alpha}, \delta = 0.01$.

by a noise which is randomly sampled from the normal distribution with mean 0 and variance $\sigma^2$. Those noisy images are used to compute bound (4).

**Results:** Table 3 reports bound (4) for $\sigma = 0.15$, ignoring the uncertainty part which is common for all models. One can observe that our bound (4) correlates very well with the test error of each model, except RegNet and VIT families. This suggests that the use of data augmentation can help us to better compare the performance of two models.

We next vary $\sigma \in \{0, 0.05, 0.1, 0.15, 0.2\}$ to see when the noise can enable a good comparison. Figure 3 reports the results about two families. We observe that while DenseNet161 has higher training error than DenseNet201 does, the error bound for DenseNet161 tends to be lower than that

**Table 3:** Bound (4) on the test error (in %) of some models which were pretrained on ImageNet dataset. Each bound was computed by adding Gaussian noises to the training images, with $\sigma = 0.15$.

| Model | Training error | Test error | Bound (4) |
|---|---|---|---|
| ResNet18 V1 | 21.245 | 30.242 | 129.226 |
| ResNet34 V1 | 15.669 | 26.686 | 111.521 |
| DenseNet161 | 10.480 | 22.862 | 94.045 |
| DenseNet169 | 12.395 | 24.400 | 100.747 |
| DenseNet201 | 9.806 | 23.104 | 96.221 |
| VGG 13 | 18.456 | 30.072 | 142.870 |
| VGG 13 BN | 19.223 | 28.414 | 134.955 |
| RegNet Y 32GF e2e | 7.127 | 13.162 | 72.474 |
| RegNet Y 32GF linear | 10.558 | 15.378 | 85.368 |
| RegNet Y 32GF V2 | 3.761 | 18.018 | 67.764 |
| VIT B 16 linear | 14.969 | 18.110 | 96.967 |
| VIT B 16 V1 | 5.916 | 18.930 | 65.969 |
| VIT L 16 linear | 11.003 | 14.850 | 80.178 |
| VIT L 16 V1 | 3.465 | 20.340 | 58.402 |

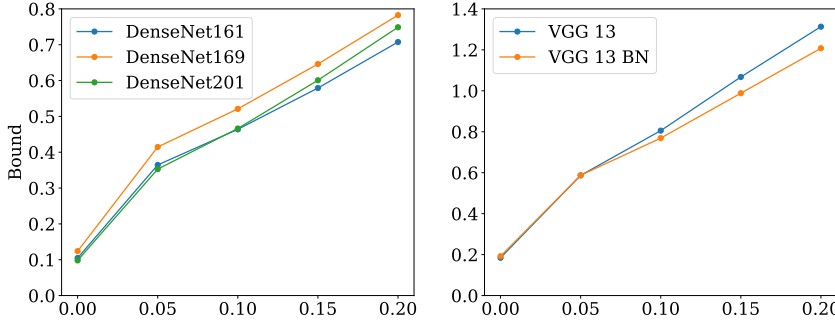

**Figure 3:** The dynamic of bound (4) as the noise level $\sigma$ increases. These subfigures report the main part $\bar{\epsilon}(\boldsymbol{h}) + F(\hat{\boldsymbol{S}}, \boldsymbol{h})$ of the bound.

of DenseNet201 as the images get more noisy. This suggests that DenseNet161 should be better than DenseNet201, which is correctly reflected by their test errors. The same behavior also appears for VGG13 and VGG13 BN. However, those two families require two different values of $\sigma$ (0.05 for VGG; 0.1 for DenseNet) to exhibit an accurate comparison. This also suggests that the anti-correlation mentioned before for RegNet and VIT may be due to the small value of $\sigma$ in Table 3. Those two families may require a higher $\sigma$ to exhibit an accurate comparison.

# 5 Conclusion

Providing theoretical guarantees for the performance of a model in practice is crucial to build reliable ML applications. Our work contributes three bounds on the test error of a model, one of which is non-vacuous for all the trained deep NNs in our experiments, without requiring any change to the trained models. Hence, our bounds can be used to provide a non-vacuous theoretical certificate for a trained model. This fills in the decade-missing cornerstone of deep learning theory.

Our work opens various avenues for future research. Indeed, while the the uncertainty part of bound (1) depends on the inherent property of the model of interest, that in bound (3) mostly does not. This suggests that bound (3) is suboptimal. One direction to develop better theories is to take more properties of a model into consideration, e.g. exploit more fine-grained properties of bound (1). Another direction is to take dependency of the training samples into account. However, it may require some improvements from very fundamental steps, e.g., concentrations for dependent variables. Since our bounds are for general settings, one interesting direction is to provide certificates for models in different types of applications, e.g. regression, segmentation, language inference, translation, text-2-images, image-2-text, ... We believe that our bounds provide a good starting point for those directions.

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

## A   Proofs for main results

*Proof of Theorem 3.1.* We first observe that

$$F(P, \boldsymbol{h}) - F(\boldsymbol{S}, \boldsymbol{h}) = F(P, \boldsymbol{h}) - \sum_{i=1}^{K} \frac{n_i}{n} a_i(\boldsymbol{h}) + \sum_{i=1}^{K} \frac{n_i}{n} a_i(\boldsymbol{h}) - F(\boldsymbol{S}, \boldsymbol{h}) \tag{5}$$

Next, we consider $F(P, \boldsymbol{h}) - \sum_{i=1}^{K} \frac{n_i}{n} a_i(\boldsymbol{h}) = \sum_{i=1}^{K} p_i a_i(\boldsymbol{h}) - \sum_{i=1}^{K} \frac{n_i}{n} a_i(\boldsymbol{h}) = \sum_{i=1}^{K} a_i(\boldsymbol{h}) \left[ p_i - \frac{n_i}{n} \right]$. Note that $(n_1, ..., n_K)$ is a multinomial random variable with parameters $n$ and $(p_1, ..., p_K)$. Therefore, according to Lemma 7 in [23], we have $\Pr\left( \sum_{i=1}^{K} a_i(\boldsymbol{h}) \left[ p_i - \frac{n_i}{n} \right] > g(\Gamma, \boldsymbol{h}, \delta_2) \right) < \delta_2$. This implies

$$\Pr\left( F(P, \boldsymbol{h}) - \sum_{i=1}^{K} \frac{n_i}{n} a_i(\boldsymbol{h}) > g(\Gamma, \boldsymbol{h}, \delta_2) \right) < \delta_2 \tag{6}$$

On the other hand, Theorem A.1 below shows that

$$\Pr\left( \sum_{i \in \boldsymbol{T}_S} \frac{n_i}{n} a_i(\boldsymbol{h}) - F(\boldsymbol{S}, \boldsymbol{h}) \geq C \sqrt{\frac{u}{2n^2} \ln \frac{1}{\delta_1}} \right) \leq \delta_1 \tag{7}$$

Combining this with (6) and the union bound, we have

$$\Pr\left( F(P, \boldsymbol{h}) > F(\boldsymbol{S}, \boldsymbol{h}) + C \sqrt{\frac{u}{2n^2} \ln \frac{1}{\delta_1}} + g(\Gamma, \boldsymbol{h}, \delta_2) \right) < \delta_1 + \delta_2 \tag{8}$$

completing the proof. □

*Proof of Theorem 3.2.* Theorem 3.1 shows that

$$\Pr\left( F(P, \boldsymbol{h}) > F(\boldsymbol{S}, \boldsymbol{h}) + C \sqrt{\frac{u}{2n^2} \ln \frac{1}{\delta_1}} + g(\Gamma, \boldsymbol{h}, \delta/2) \right) < \delta_1 + \delta/2 \tag{9}$$

where $u$ and $\delta_1$ depend on the sum $\sum_{i=1}^{K} p_i^2$. We next bound this quantity using $\boldsymbol{S}$.

Since $p_i \geq 0$ and $\sum_{i=1}^{K} p_i = 1$, we can use the Lagrange multiplier method to show that $\sum_{i=1}^{K} p_i^2$ is minimized at $1/K$. Hence $u = \sum_{i=1}^{K} \gamma n p_i (1 + \gamma n p_i) = \gamma n + \gamma^2 n^2 \sum_{i=1}^{K} p_i^2 \geq \gamma n + \gamma^2 n^2 / K$. This suggests that $\exp(-\frac{u \ln \gamma}{4n-3}) \leq \exp(-\frac{(\gamma n + \gamma^2 n^2/K) \ln \gamma}{4n-3}) \leq \exp(-\frac{\gamma n(K+\gamma n) \ln \gamma}{K(4n-3)}) \leq \gamma^{-\alpha}$. Choosing $\delta_1 = \gamma^{-\alpha}$ and plugging it into (9) lead to

$$\Pr\left( F(P, \boldsymbol{h}) > F(\boldsymbol{S}, \boldsymbol{h}) + C \sqrt{\frac{u}{2n^2} \alpha \ln \gamma} + g(\Gamma, \boldsymbol{h}, \delta/2) \right) < \delta/2 + \gamma^{-\alpha} \tag{10}$$

It is easy to see that $g(\Gamma, \boldsymbol{h}, \delta/2) \leq g_2(\delta/2)$, since $a_o(\boldsymbol{h}) \leq C$ and $a_i(\boldsymbol{h}) \leq C$ for any $i$. Therefore

$$\Pr\left( F(P, \boldsymbol{h}) > F(\boldsymbol{S}, \boldsymbol{h}) + C \sqrt{\frac{u}{2n^2} \alpha \ln \gamma} + g_2(\delta/2) \right) < \delta/2 + \gamma^{-\alpha} \tag{11}$$

Next we consider $\frac{u}{2n^2} = \frac{\gamma}{2n} + \frac{\gamma^2}{2} \sum_{i=1}^{K} p_i^2$. Since $\boldsymbol{S}$ contains $n$ i.i.d. samples, $(n_1, ..., n_K)$ is a multinomial random variable with parameters $n$ and $(p_1, ..., p_K)$. Lemma B.8 shows

$$\Pr\left( \sum_{i=1}^{K} p_i^2 > \sum_{i=1}^{K} \left( \frac{n_i}{n} \right)^2 + 2 \sqrt{\frac{2}{n} \ln \frac{2K}{\delta}} \right) < \delta/2$$

Therefore $\Pr\left( \frac{u}{2n^2} > \frac{\gamma}{2n} + \frac{\gamma^2}{2} \sum_{i=1}^{K} \left( \frac{n_i}{n} \right)^2 + \gamma^2 \sqrt{\frac{2}{n} \ln \frac{2K}{\delta}} \right) < \delta/2$. This also suggests that

$$\Pr\left( C \sqrt{\frac{u}{2n^2} \alpha \ln \gamma} > C \sqrt{\hat{u} \alpha \ln \gamma} \right) < \delta/2 \tag{12}$$

Combining this with (11) and the union bound will complete the proof. □

*Proof of Theorem 3.3.* Theorem 3.2 shows that the following holds with probability at least $1 - \gamma^{-\alpha} - \delta$:

$$F(P, \boldsymbol{h}) \leq F(\boldsymbol{S}, \boldsymbol{h}) + C\sqrt{\hat{u}\alpha \ln \gamma} + g(\delta/2) \tag{13}$$

Note that

$$F(\boldsymbol{S}, \boldsymbol{h}) = F(\boldsymbol{S}, \boldsymbol{h}) - \sum_{i \in \boldsymbol{T}} \frac{m_i}{m} F(\boldsymbol{S}_i, \boldsymbol{h}) + \sum_{i \in \boldsymbol{T}} \frac{m_i}{m} F(\boldsymbol{S}_i, \boldsymbol{h}) - F(\hat{\boldsymbol{S}}, \boldsymbol{h}) + F(\hat{\boldsymbol{S}}, \boldsymbol{h}) \tag{14}$$

$$= \sum_{i \in \boldsymbol{T}} \frac{m_i}{m} [F(\boldsymbol{S}_i, \boldsymbol{h}) - F(\hat{\boldsymbol{S}}_i, \boldsymbol{h})] + F(\boldsymbol{S}, \boldsymbol{h}) - \sum_{i \in \boldsymbol{T}} \frac{m_i}{m} F(\boldsymbol{S}_i, \boldsymbol{h}) + F(\hat{\boldsymbol{S}}, \boldsymbol{h}) \tag{15}$$

$$= \sum_{i \in \boldsymbol{T}} \frac{m_i}{m} \frac{1}{n_i} \sum_{\boldsymbol{z} \in \boldsymbol{S}_i} [\ell(\boldsymbol{h}, \boldsymbol{z}) - F(\hat{\boldsymbol{S}}_i, \boldsymbol{h})] + F(\boldsymbol{S}, \boldsymbol{h}) - \sum_{i \in \boldsymbol{T}} \frac{m_i}{m} F(\boldsymbol{S}_i, \boldsymbol{h}) + F(\hat{\boldsymbol{S}}, \boldsymbol{h}) \tag{16}$$

$$= \sum_{i \in \boldsymbol{T}} \frac{m_i}{m} \frac{1}{n_i m_i} \sum_{\boldsymbol{z} \in \boldsymbol{S}_i, \boldsymbol{s} \in \hat{\boldsymbol{S}}_i} [\ell(\boldsymbol{h}, \boldsymbol{z}) - \ell(\boldsymbol{h}, \boldsymbol{s})] + F(\boldsymbol{S}, \boldsymbol{h}) - \sum_{i \in \boldsymbol{T}} \frac{m_i}{m} F(\boldsymbol{S}_i, \boldsymbol{h}) + F(\hat{\boldsymbol{S}}, \boldsymbol{h}) \tag{17}$$

$$\leq \sum_{i \in \boldsymbol{T}} \frac{m_i}{m} \frac{1}{n_i m_i} \sum_{\boldsymbol{z} \in \boldsymbol{S}_i, \boldsymbol{s} \in \hat{\boldsymbol{S}}_i} |\ell(\boldsymbol{h}, \boldsymbol{z}) - \ell(\boldsymbol{h}, \boldsymbol{s})| + F(\boldsymbol{S}, \boldsymbol{h}) - \sum_{i \in \boldsymbol{T}} \frac{m_i}{m} F(\boldsymbol{S}_i, \boldsymbol{h}) + F(\hat{\boldsymbol{S}}, \boldsymbol{h}) \tag{18}$$

$$\leq \sum_{i \in \boldsymbol{T}} \frac{m_i}{m} \bar{\epsilon}_i + F(\boldsymbol{S}, \boldsymbol{h}) - \sum_{i \in \boldsymbol{T}} \frac{m_i}{m} F(\boldsymbol{S}_i, \boldsymbol{h}) + F(\hat{\boldsymbol{S}}, \boldsymbol{h}) \tag{19}$$

$$= \sum_{i \in \boldsymbol{T}} \frac{m_i}{m} \bar{\epsilon}_i + \sum_{i \in \boldsymbol{T}} \left( \frac{n_i}{n} - \frac{m_i}{m} \right) F(\boldsymbol{S}_i, \boldsymbol{h}) + F(\hat{\boldsymbol{S}}, \boldsymbol{h}) \tag{20}$$

Since this determistically holds for all $\boldsymbol{S}$, combining (13) with (20) completes the proof. □

## A.1 Approximating the intractable part by a data set

**Theorem A.1.** *Given the notations in Theorem 3.1,*

$$\Pr \left( \sum_{i \in \boldsymbol{T}_S} \frac{n_i}{n} a_i(\boldsymbol{h}) \geq \sum_{i \in \boldsymbol{T}_S} \frac{n_i}{n} F(\boldsymbol{S}_i, \boldsymbol{h}) + C\sqrt{\frac{u}{2n^2} \ln \frac{1}{\delta_1}} \right) \leq \delta_1 \tag{21}$$

*Proof.* Denote $\boldsymbol{n} = \{n_1, ..., n_K\}$ and for each $j \in [K]$:

$$B_j = \sum_{i=1}^{j} n_i a_i(\boldsymbol{h}) - \sum_{i=1}^{j} n_i F(\boldsymbol{S}_i, \boldsymbol{h}) \tag{22}$$

$$X_j = n_j F(\boldsymbol{S}_j, \boldsymbol{h}) \tag{23}$$

$$\boldsymbol{S}_{\leq j} = \bigcup_{i \leq j} \boldsymbol{S}_i \tag{24}$$

Denote $y = \frac{4t}{uC^2}$ for any $t \in \left[ 0, uC\sqrt{\frac{\ln \gamma}{8n-6}} \right]$. The proof for (21) contains three main steps.

**Step 1:** We first observe that

$$\Pr(B_K \geq t) \leq e^{-yt} \mathbb{E}_{\boldsymbol{S}} \left[ e^{yB_K} \right] \qquad \text{(Chernoff bounds)} \tag{25}$$

$$\leq e^{-yt} \mathbb{E}_{\boldsymbol{h}, \boldsymbol{n}} \left[ \mathbb{E}_{\boldsymbol{S}} \left[ e^{yB_K} | \boldsymbol{h}, \boldsymbol{n} \right] \right] \qquad \text{(Law of total expectation)} \tag{26}$$

**Step 2 - estimating $\mathbb{E}_{\boldsymbol{S}}\left[e^{yB_K}|\boldsymbol{h},\boldsymbol{n}\right]$:** We observe the following for each $j \in \boldsymbol{T}_S$,

$$\mathbb{E}_{X_j}[X_j|\boldsymbol{h},\boldsymbol{n}] = \mathbb{E}_{\boldsymbol{S}_j}[n_j F(\boldsymbol{S}_j,\boldsymbol{h})|\boldsymbol{h},\boldsymbol{n}] \tag{27}$$

$$= \mathbb{E}_{\boldsymbol{S}_j}\left[\sum_{i=1}^{n_j}\ell(\boldsymbol{h},\boldsymbol{z}_{ji})|\boldsymbol{h},\boldsymbol{n}\right] \qquad \text{(where } \boldsymbol{S}_j = \{\boldsymbol{z}_{ji}\}_{i=1}^{n_j}) \tag{28}$$

$$= \sum_{i=1}^{n_j}\mathbb{E}_{\boldsymbol{z}_{ji}\in\mathcal{Z}_j}\left[\ell(\boldsymbol{h},\boldsymbol{z}_{ji})|\boldsymbol{h},\boldsymbol{n}\right] \qquad (\boldsymbol{S}_j \text{ contains i.i.d. samples in } \mathcal{Z}_j) \tag{29}$$

$$= \sum_{i=1}^{n_j} a_j(\boldsymbol{h}) = n_j a_j(\boldsymbol{h}) \tag{30}$$

Therefore $B_j = B_{j-1} + \mathbb{E}_{X_j}[X_j|\boldsymbol{h},\boldsymbol{n}] - X_j$ for all $j \in \boldsymbol{T}_S$. Note that $B_i = B_{i-1}$ (due to $n_i = b_i = X_i = 0$) for all $i \notin \boldsymbol{T}_S$. Hence, for $i \notin \boldsymbol{T}_S$, we will use $\mathbb{E}_{X_i}[X_i|\boldsymbol{h},\boldsymbol{n}] - X_i$ instead of 0 in the below analysis for simplicity of presentation.

We can rewrite

$$\mathbb{E}_{\boldsymbol{S}}\left[e^{yB_K}|\boldsymbol{h},\boldsymbol{n}\right] = \mathbb{E}_{\boldsymbol{S}}\left[e^{y(B_{K-1}+\mathbb{E}_{X_K}[X_K|\boldsymbol{h},\boldsymbol{n}]-X_K)}|\boldsymbol{h},\boldsymbol{n}\right] \tag{31}$$

$$= \mathbb{E}_{\boldsymbol{S}_{\leq K}}\left[e^{y(B_{K-1}+\mathbb{E}_{X_K}[X_K|\boldsymbol{h},\boldsymbol{n}]-X_K)}|\boldsymbol{h},\boldsymbol{n}\right] \tag{32}$$

$$\leq \mathbb{E}_{\boldsymbol{S}_{\leq K-1}}\left[e^{yB_{K-1}}|\boldsymbol{h},\boldsymbol{n}\right]\mathbb{E}_{X_K}\left[e^{y(\mathbb{E}_{X_K}[X_K|\boldsymbol{h},\boldsymbol{n}]-X_K)}|\boldsymbol{h},\boldsymbol{n}\right] \tag{33}$$

where the last inequality comes from the fact that $X_K$ is conditionally independent with $\boldsymbol{S}_{\leq K-1}$, conditioned on $\{\boldsymbol{h},\boldsymbol{n}\}$.

It is easy to see that $0 \leq X_K \leq Cn_K$, due to $0 \leq F(\boldsymbol{S}_K,\boldsymbol{h}) \leq C$. Lemma B.1 implies $\mathbb{E}_{X_K}\left[e^{y(\mathbb{E}_{X_K}[X_K|\boldsymbol{h},\boldsymbol{n}]-X_K)}|\boldsymbol{h},\boldsymbol{n}\right] \leq \exp\left(\frac{y^2C^2n_K^2}{8}\right)$. Plugging this into (33), we obtain

$$\mathbb{E}_{\boldsymbol{S}}\left[e^{yB_K}|\boldsymbol{h},\boldsymbol{n}\right] \leq \mathbb{E}_{\boldsymbol{S}_{\leq K-1}}\left[e^{yB_{K-1}}|\boldsymbol{h},\boldsymbol{n}\right]\exp\left(\frac{y^2C^2n_K^2}{8}\right) \tag{34}$$

Using the same arguments for $X_{K-1}, ..., X_1$, we obtain the followings

$$\mathbb{E}_{\boldsymbol{S}}\left[e^{yB_K}|\boldsymbol{h},\boldsymbol{n}\right] \leq \mathbb{E}_{\boldsymbol{S}_{\leq K-2}}\left[e^{yB_{K-2}}|\boldsymbol{h},\boldsymbol{n}\right]\exp\left(\frac{y^2C^2n_K^2}{8} + \frac{y^2C^2n_{K-1}^2}{8}\right)$$

$$...$$

$$\leq \exp\left(\frac{y^2C^2}{8}\sum_{i=1}^{K}n_i^2\right) \tag{35}$$

**Step 3 - bounding $\Pr\left(B_K \geq t\right)$:** By combining this with (26), we obtain

$$\Pr\left(B_K \geq t\right) \leq e^{-yt}\mathbb{E}_{\boldsymbol{h},\boldsymbol{n}}\exp\left(\frac{y^2C^2}{8}\sum_{i=1}^{K}n_i^2\right) \tag{36}$$

$$= e^{-yt}\mathbb{E}_{\boldsymbol{n}}\exp\left(\frac{y^2C^2}{8}\sum_{i=1}^{K}n_i^2\right) \tag{37}$$

$$\leq e^{-yt}\mathbb{E}_{\boldsymbol{n}}\exp\left(\frac{y^2C^2}{8}\sum_{i=1}^{K-1}n_i^2\right)\mathbb{E}_{n_K}\exp\left(\frac{y^2C^2}{8}n_K^2\right) \tag{38}$$

$$(\text{Since } n_K \text{ is independent with } v_1, ..., n_{K-1})$$

When $\gamma p_K < 1$, due to $t \leq uC\sqrt{\frac{\ln\gamma}{8n-6}}$, observe that $\frac{y^2C^2}{8} = \frac{2t^2}{u^2C^2} \leq \frac{\ln\gamma}{4n-3} \leq \frac{\ln\gamma}{(1-\gamma p_K)(4n-3)}$. Note that $n_K$ is a binomial random variable with parameters $n$ and $p_K$. Combining those facts with Lemma B.7 implies $\mathbb{E}_{n_K}\exp\left(\frac{y^2C^2}{8}n_K^2\right) \leq \exp\left(\frac{y^2C^2}{8}\gamma np_K(1+\gamma np_K)\right)$. On the other hand, Lemma B.6

also implies $\mathbb{E}_{n_K} \exp\left(\frac{y^2 C^2}{8} n_K^2\right) \leq \exp\left(\frac{y^2 C^2}{8} \gamma n p_K(1 + \gamma n p_K)\right)$ when $\gamma p_K \geq 1$. As a result, those facts and (38) lead to the following:

$$\Pr\left(B_K \geq t\right) \quad \leq \quad e^{-yt}\mathbb{E}_{\boldsymbol{n}} \exp\left(\frac{y^2 C^2}{8}\sum_{i=1}^{K-1} n_i^2\right) \exp\left(\frac{y^2 C^2}{8}\left((1 + \gamma n p_K)\gamma n p_K\right)\right) \quad (39)$$

Using the same arguments for the remaining variables $n_{K-1}, ..., n_1$, we obtain

$$\Pr\left(B_K \geq t\right) \quad \leq \quad \exp\left(-yt + \frac{y^2 C^2}{8}\sum_{i=1}^{K}(1 + \gamma n p_i)\gamma n p_i\right) \quad (40)$$

$$= \quad \exp\left(-yt + \frac{y^2 C^2 u}{8}\right) = \exp\left(\frac{-2t^2}{uC^2}\right) \quad (41)$$

As a result

$$\Pr\left(\sum_{i=1}^{K} n_i a_i(\boldsymbol{h}) \geq \sum_{i=1}^{K} n_i F(\boldsymbol{S}_i, \boldsymbol{h}) + t\right) \quad \leq \quad \exp\left(-\frac{2t^2}{uC^2}\right) \quad (42)$$

Since $n_j = 0$ for all $j \notin \boldsymbol{T}_S$, we have

$$\Pr\left(\sum_{i \in \boldsymbol{T}_S} n_i a_i(\boldsymbol{h}) \geq \sum_{i \in \boldsymbol{T}_S} n_i F(\boldsymbol{S}_i, \boldsymbol{h}) + t\right) \quad \leq \quad \exp\left(-\frac{2t^2}{uC^2}\right) \quad (43)$$

Multiplying both sides (of the probability term) with $1/n$ leads to

$$\Pr\left(\sum_{i \in \boldsymbol{T}_S} \frac{n_i}{n} a_i(\boldsymbol{h}) \geq \sum_{i \in \boldsymbol{T}_S} \frac{n_i}{n} F(\boldsymbol{S}_i, \boldsymbol{h}) + t/n\right) \leq \exp\left(-\frac{2t^2}{uC^2}\right)$$

Choosing $t = C\sqrt{\frac{u}{2} \ln\frac{1}{\delta_1}}$ results in (21), completing the proof. $\qquad\square$

# B   Supporting theorems and lemmas

## B.1   Hoeffding's Lemma

**Lemma B.1** (Hoeffding's lemma for conditionals). *Let $X$ be any real-valued random variable that may depend on some random variables $\boldsymbol{Y}$. Assume that $a \leq X \leq b$ almost surely, for some constants $a, b$. Then, for all $\lambda \in \mathbb{R}$,*

$$\mathbb{E}_X\left[e^{\lambda(\mathbb{E}_X[X|\boldsymbol{Y}]-X)}|\boldsymbol{Y}\right] \quad \leq \quad \exp\left(\frac{\lambda^2(b-a)^2}{8}\right) \quad (44)$$

*Proof.* Denote $c = \mathbb{E}_X[X|\boldsymbol{Y}] - b, d = \mathbb{E}_X[X|\boldsymbol{Y}] - a$ and hence $c \leq 0 \leq d$.

Since $\exp$ is a convex function, we have the following for all $\mathbb{E}_X[X|\boldsymbol{Y}] - X \in [c, d]$:

$$e^{\lambda(\mathbb{E}_X[X|\boldsymbol{Y}]-X)} \leq \frac{d - \mathbb{E}_X[X|\boldsymbol{Y}] + X}{d-c}e^{\lambda c} + \frac{\mathbb{E}_X[X|\boldsymbol{Y}] - X - c}{d-c}e^{\lambda d}$$

Therefore, by taking the conditional expectation over $X$ for both sides,

$$\mathbb{E}_X\left[e^{\lambda(\mathbb{E}_X[X|\boldsymbol{Y}]-X)}|\boldsymbol{Y}\right] \quad \leq \quad \frac{d - \mathbb{E}_X[X|\boldsymbol{Y}] + \mathbb{E}_X[X|\boldsymbol{Y}]}{d-c}e^{\lambda c} + \frac{\mathbb{E}_X[X|\boldsymbol{Y}] - \mathbb{E}_X[X|\boldsymbol{Y}] - c}{d-c}e^{\lambda d}$$

$$= \quad \frac{d}{d-c}e^{\lambda c} - \frac{c}{d-c}e^{\lambda d} \quad (45)$$

$$= \quad e^{L(\lambda(d-c))} \quad (46)$$

where $L(h) = \frac{ch}{d-c} + \ln(1 + \frac{c - e^h c}{d-c})$. For this function, note that

$$L(0) = L'(0) = 0 \text{ and } L''(h) = -\frac{cde^h}{(d - ce^h)^2}$$

The AM-GM inequality suggests that $L''(h) \leq 1/4$ for all $h$. Combining this property with Taylor's theorem leads to the following, for some $\theta \in [0, 1]$,

$$L(h) = L(0) + hL'(0) + \frac{1}{2}h^2 L''(h\theta) \leq \frac{h^2}{8}$$

Combining this with (46) completes the proof. □

## B.2 Small random variables

**Lemma B.2.** *Let $x_1, ..., x_n$ be independent random variables in $[0, 1]$ and satisfy $\mathbb{E}[x_i] \leq \nu, \forall i$ for some $\nu \in [0, 1]$. For any $c \geq 1$ satisfying $c\nu \geq 1$ and any $\lambda \geq 0$, we have $\mathbb{E}\exp\left(\lambda(x_1 + \cdots + x_n)^2\right) \leq \exp(\lambda cn\nu(1 + cn\nu))$.*

**Lemma B.3.** *Let $x_1, ..., x_n$ be independent random variables in $[0, 1]$ and satisfy $\mathbb{E}[x_i] \leq \nu, \forall i$ for some $\nu \in [0, 1]$. For any $c \geq 1$ satisfying $c\nu < 1$ and any $\lambda \in [0, \frac{\ln c}{(1-c\nu)(4n-3)}]$, we have $\mathbb{E}\exp\left(\lambda(x_1 + \cdots + x_n)^2\right) \leq \exp(\lambda cn\nu(1 + cn\nu))$.*

In order to prove those results, we need the following observations.

**Lemma B.4.** *Consider a random variable $X \in [0, 1]$ with mean $\mathbb{E}[X] \leq \nu$ for some constant $\nu \in [0, 1]$. For any $c \geq 1, \lambda \geq 0$:*

- *If $c\nu \geq 1$, then $\mathbb{E}e^{\lambda X} \leq e^{c\nu\lambda}$.*
- *If $c\nu < 1$, then $\mathbb{E}e^{\lambda X} \leq e^{c\nu\lambda}$ for all $\lambda \in [0, \frac{\ln c}{1-c\nu}]$.*

*Proof.* The Taylor series expansion of the function $e^{\lambda X}$ at any $X$ is $e^{\lambda X} = 1 + \sum_{p=1}^{\infty} \frac{(\lambda X)^p}{p!}$. Therefore

$$\mathbb{E}[e^{\lambda X}] = 1 + \sum_{p=1}^{\infty} \frac{\lambda^p}{p!}\mathbb{E}(X^p) \leq 1 + \mathbb{E}(X)\sum_{p=1}^{\infty} \frac{\lambda^p}{p!} \quad \text{(due to } X^p \leq X, \forall p \geq 1\text{)} \quad (47)$$

$$\leq 1 + \nu\sum_{p=1}^{\infty} \frac{\lambda^p}{p!} = 1 + \nu(e^\lambda - 1) = 1 - \nu + \nu e^\lambda \quad (48)$$

Next we consider function $y(\lambda) = e^{c\nu\lambda} - 1 + \nu - \nu e^\lambda$. Its derivative is $y' = c\nu e^{c\nu\lambda} - \nu e^\lambda = \nu e^\lambda(ce^{(c\nu-1)\lambda} - 1)$.

For the case $c\nu \geq 1$, one can observe that $y' \geq 0$ for all $\lambda \geq 0$. This means $y$ is non-decreasing, and hence $y(\lambda) \geq y(0) = 0$. As a result, $e^{c\nu\lambda} \geq 1 - \nu + \nu e^\lambda \geq \mathbb{E}[e^{\lambda X}]$.

Consider the case $c\nu < 1$, it is easy to show that $y'(\lambda) \geq 0$ for all $\lambda \in [0, \frac{\ln c}{1-c\nu}]$. This means $y$ is non-decreasing in the interval $[0, \frac{\ln c}{1-c\nu}]$, and hence $y(\lambda) \geq y(0) = 0$ for all $\lambda \in [0, \frac{\ln c}{1-c\nu}]$. As a result, $e^{c\nu\lambda} \geq 1 - \nu + \nu e^\lambda \geq \mathbb{E}[e^{\lambda X}]$, completing the proof. □

**Corollary B.5.** *Consider a random variable $X \in [0, 1]$ with mean $\mathbb{E}[X] \leq \nu$ for some constant $\nu \in [0, 1]$. For all constants $a, b \geq 0, c \geq 1$:*

- *$\mathbb{E}e^{\lambda(aX^2+bX)} \leq e^{c(a+b)\nu\lambda}$, for all $\lambda \geq 0$, if $c\nu \geq 1$.*
- *$\mathbb{E}e^{\lambda(aX^2+bX)} \leq e^{c(a+b)\nu\lambda}$, for all $\lambda \in [0, \frac{\ln c}{(1-c\nu)(a+b)}]$, if $c\nu < 1$.*

*Proof.* It is easy to observe that $\mathbb{E}e^{\lambda(aX^2)} \leq \mathbb{E}e^{\lambda(aX)}$ due to $X \in [0, 1]$. This suggests that $\mathbb{E}e^{\lambda(aX^2+bX)} \leq \mathbb{E}e^{\lambda(a+b)X}$. Applying Lemma B.4 will complete the proof. □

*Proof of Lemma B.2.* Denote $y_n = x_1 + \cdots + x_n$. Observe that $y_n = y_{n-1} + x_n$ and

$$\mathbb{E}_{y_n} e^{\lambda y_n^2} = \mathbb{E}_{y_n} e^{\lambda(y_{n-1}^2 + 2x_n y_{n-1} + x_n^2)} = \mathbb{E}_{y_{n-1}} \left[ e^{\lambda y_{n-1}^2} \mathbb{E}_{x_n} e^{\lambda(2x_n y_{n-1} + x_n^2)} \right] \quad (49)$$

Since $c\nu \geq 1$ and $x_n$ is independent with $y_{n-1}$, Corollary B.5 implies $\mathbb{E}_{x_n} e^{\lambda(2x_n y_{n-1} + x_n^2)} \leq e^{c\nu\lambda(2y_{n-1}+1)}$. Plugging this into (49) leads to

$$\mathbb{E}_{y_n} e^{\lambda y_n^2} \leq \mathbb{E}_{y_{n-1}} \left[ e^{\lambda y_{n-1}^2} e^{c\nu\lambda(2y_{n-1}+1)} \right] = e^{c\nu\lambda} \mathbb{E}_{y_{n-1}} \left[ e^{\lambda(y_{n-1}^2 + 2c\nu y_{n-1})} \right] \quad (50)$$

Next we consider $\mathbb{E}_{y_{n-1}} \left[ e^{\lambda(y_{n-1}^2 + 2c\nu y_{n-1})} \right]$. Observe that $y_{n-1} = y_{n-2} + x_{n-1}$ and hence

$$\mathbb{E}_{y_{n-1}} \left[ e^{\lambda(y_{n-1}^2 + 2c\nu y_{n-1})} \right] = \mathbb{E}_{y_{n-1}} e^{\lambda(y_{n-2}^2 + 2x_{n-1}y_{n-2} + x_{n-1}^2 + 2c\nu x_{n-1} + 2c\nu y_{n-2})} \quad (51)$$

$$= \mathbb{E}_{y_{n-2}} \left[ e^{\lambda(y_{n-2}^2 + 2c\nu y_{n-2})} \mathbb{E}_{x_{n-1}} e^{\lambda(2x_{n-1}y_{n-2} + 2c\nu x_{n-1} + x_{n-1}^2)} \right] \quad (52)$$

Since $c\nu \geq 1$ and $x_{n-1}$ is independent with $y_{n-2}$, Corollary B.5 implies $\mathbb{E}_{x_{n-1}} e^{\lambda(2x_{n-1}y_{n-2} + 2c\nu x_{n-1} + x_{n-1}^2)} \leq e^{c\nu\lambda(2y_{n-2} + 2c\nu + 1)}$. Plugging this into (52) leads to

$$\mathbb{E}_{y_{n-1}} \left[ e^{\lambda(y_{n-1}^2 + 2c\nu y_{n-1})} \right] \leq \mathbb{E}_{y_{n-2}} \left[ e^{\lambda(y_{n-2}^2 + 2c\nu y_{n-2})} e^{c\nu\lambda(2y_{n-2} + 2c\nu + 1)} \right] \quad (53)$$

$$= e^{c\nu\lambda(2c\nu+1)} \mathbb{E}_{y_{n-2}} \left[ e^{\lambda(y_{n-2}^2 + 4c\nu y_{n-2})} \right] \quad (54)$$

By using the same arguments, we can show that

$$\mathbb{E}_{y_{n-1}} \left[ e^{\lambda(y_{n-1}^2 + 2c\nu y_{n-1})} \right] \leq e^{c\nu\lambda(2c\nu+1)} e^{c\nu\lambda(4c\nu+1)} \mathbb{E}_{y_{n-3}} \left[ e^{\lambda(y_{n-3}^2 + 6c\nu y_{n-3})} \right] \quad (55)$$

$$= e^{2c\nu\lambda(3c\nu+1)} \mathbb{E}_{y_{n-3}} \left[ e^{\lambda(y_{n-3}^2 + 6c\nu y_{n-3})} \right] \quad (56)$$

$$...$$

$$\leq e^{c(n-2)\nu\lambda(c(n-1)\nu+1)} \mathbb{E}_{y_1} \left[ e^{\lambda(y_1^2 + 2c(n-1)\nu y_1)} \right] \quad (57)$$

Note that $\mathbb{E}_{y_1} \left[ e^{\lambda(y_1^2 + 2c(n-1)\nu y_1)} \right] = \mathbb{E}_{x_1} \left[ e^{\lambda(x_1^2 + 2c(n-1)\nu x_1)} \right] \leq e^{c\nu\lambda(1+2c(n-1)\nu)}$, according to Corollary B.5. Combining this with (57), we obtain

$$\mathbb{E}_{y_{n-1}} \left[ e^{\lambda(y_{n-1}^2 + 2c\nu y_{n-1})} \right] \leq e^{c(n-2)\nu\lambda(c(n-1)\nu+1)} e^{c\nu\lambda(1+2c(n-1)\nu)} = e^{c\nu\lambda(1+cn\nu)(n-1)} \quad (58)$$

By plugging this into (50), we obtain

$$\mathbb{E}_{y_n} e^{\lambda y_n^2} \leq e^{c\nu\lambda} e^{c\nu\lambda(1+cn\nu)(n-1)} = e^{c\nu\lambda((1+cn\nu)n - cn\nu)} \quad (59)$$

$$\leq e^{cn\nu(1+cn\nu)\lambda} \quad (60)$$

completing the proof. $\square$

*Proof of Lemma B.3.* Denote $y_n = x_1 + \cdots + x_n$ and observe that

$$\mathbb{E}_{y_n} e^{\lambda y_n^2} = \mathbb{E}_{y_n} e^{\lambda(y_{n-1}^2 + 2x_n y_{n-1} + x_n^2)} = \mathbb{E}_{y_{n-1}} \left[ e^{\lambda y_{n-1}^2} \mathbb{E}_{x_n} e^{\lambda(2x_n y_{n-1} + x_n^2)} \right] \quad (61)$$

Note that $y_{n-1} = x_1 + \cdots + x_{n-1} \leq n - 1$ and $\lambda(2y_{n-1} + 1) \leq \lambda(2n - 1) \leq \lambda(4n - 3) \leq \frac{\ln c}{1 - c\nu}$. Since $x_n$ is independent with $y_{n-1}$, Corollary B.5 implies $\mathbb{E}_{x_n} e^{\lambda(2x_n y_{n-1} + x_n^2)} \leq e^{c\nu\lambda(2y_{n-1}+1)}$. Plugging this into (61) leads to

$$\mathbb{E}_{y_n} e^{\lambda y_n^2} \leq \mathbb{E}_{y_{n-1}} \left[ e^{\lambda y_{n-1}^2} e^{c\nu\lambda(2y_{n-1}+1)} \right] = e^{c\nu\lambda} \mathbb{E}_{y_{n-1}} \left[ e^{\lambda(y_{n-1}^2 + 2c\nu y_{n-1})} \right] \quad (62)$$

Next we consider $\mathbb{E}_{y_{n-1}} \left[ e^{\lambda(y_{n-1}^2 + 2c\nu y_{n-1})} \right]$. Observe that

$$\mathbb{E}_{y_{n-1}}\left[e^{\lambda(y_{n-1}^2+2c\nu y_{n-1})}\right] \quad = \quad \mathbb{E}_{y_{n-1}}e^{\lambda(y_{n-2}^2+2x_{n-1}y_{n-2}+x_{n-1}^2+2c\nu x_{n-1}+2c\nu y_{n-2})} \tag{63}$$

$$= \quad \mathbb{E}_{y_{n-2}}\left[e^{\lambda(y_{n-2}^2+2c\nu y_{n-2})}\mathbb{E}_{x_{n-1}}e^{\lambda(2x_{n-1}y_{n-2}+2c\nu x_{n-1}+x_{n-1}^2)}\right] \tag{64}$$

One can easily show that $\lambda(2y_{n-2}+2c\nu+1) \le \lambda(2(n-2)+2c\nu+1) \le \lambda(4n-3) \le \frac{\ln c}{1-c\nu}$, since $y_{n-2} = x_1 + \cdots + x_{n-2} \le n-2$. Therefore Corollary B.5 implies $\mathbb{E}_{x_{n-1}}e^{\lambda(2x_{n-1}y_{n-2}+2c\nu x_{n-1}+x_{n-1}^2)} \le e^{c\nu\lambda(2y_{n-2}+2c\nu+1)}$, since $x_{n-1}$ is independent with $y_{n-2}$. Plugging this into (64) leads to

$$\mathbb{E}_{y_{n-1}}\left[e^{\lambda(y_{n-1}^2+2c\nu y_{n-1})}\right] \quad \le \quad \mathbb{E}_{y_{n-2}}\left[e^{\lambda(y_{n-2}^2+2c\nu y_{n-2})}e^{c\nu\lambda(2y_{n-2}+2c\nu+1)}\right] \tag{65}$$

$$= \quad e^{c\nu\lambda(2c\nu+1)}\mathbb{E}_{y_{n-2}}\left[e^{\lambda(y_{n-2}^2+4c\nu y_{n-2})}\right] \tag{66}$$

By using the same arguments, we can show that

$$\mathbb{E}_{y_{n-1}}\left[e^{\lambda(y_{n-1}^2+2c\nu y_{n-1})}\right] \quad \le \quad e^{c\nu\lambda(2c\nu+1)}e^{c\nu\lambda(4c\nu+1)}\mathbb{E}_{y_{n-3}}\left[e^{\lambda(y_{n-3}^2+6c\nu y_{n-3})}\right] \tag{67}$$

$$= \quad e^{2c\nu\lambda(3c\nu+1)}\mathbb{E}_{y_{n-3}}\left[e^{\lambda(y_{n-3}^2+6c\nu y_{n-3})}\right] \tag{68}$$

$$...$$

$$\le \quad e^{c(n-2)\nu\lambda(c(n-1)\nu+1)}\mathbb{E}_{y_1}\left[e^{\lambda(y_1^2+2c(n-1)\nu y_1)}\right] \tag{69}$$

Note that $\mathbb{E}_{y_1}\left[e^{\lambda(y_1^2+2c(n-1)\nu y_1)}\right] = \mathbb{E}_{x_1}\left[e^{\lambda(x_1^2+2c(n-1)\nu x_1)}\right] \le e^{c\nu\lambda(1+2c(n-1)\nu)}$, according to Corollary B.5 and the fact that $\lambda(1+2c(n-1)\nu) \le \lambda(4n-3) \le \frac{\ln c}{1-c\nu}$. Combining this with (69), we obtain

$$\mathbb{E}_{y_{n-1}}\left[e^{\lambda(y_{n-1}^2+2c\nu y_{n-1})}\right] \quad \le \quad e^{c(n-2)\nu\lambda(c(n-1)\nu+1)}e^{c\nu\lambda(1+2c(n-1)\nu)} = e^{c\nu\lambda(1+cn\nu)(n-1)} \tag{70}$$

By plugging this into (62), we obtain

$$\mathbb{E}_{y_n}e^{\lambda y_n^2} \quad \le \quad e^{c\nu\lambda}e^{c\nu\lambda(1+cn\nu)(n-1)} = e^{c\nu\lambda((1+cn\nu)n-cn\nu)} \tag{71}$$

$$\le \quad e^{cn\nu(1+cn\nu)\lambda} \tag{72}$$

completing the proof. $\qquad\square$

## B.3 Binomial and multinomial random variables

Next we analyze some properties of binomial random variables.

**Lemma B.6.** *Consider a binomial random variable $z$ with parameters $n \ge 1$ and $\nu \in [0,1]$. For any $c \ge 1$ satisfying $c\nu \ge 1$ and any $\lambda \ge 0$, we have $\mathbb{E}e^{\lambda z^2} \le e^{cn\nu(1+cn\nu)\lambda}$.*

*Proof.* Since $z$ is a binomial random variable, we can write $z = x_1 + \cdots + x_n$, where $x_1, ..., x_n$ are i.i.d. Bernoulli random variables with parameter $\nu$. Therefore applying Lemma B.2 completes the proof. $\qquad\square$

**Lemma B.7.** *Consider a binomial random variable $z$ with parameters $n \ge 1$ and $\nu \in [0,1]$. For any $c \ge 1$ satisfying $c\nu < 1$ and any $\lambda \in [0, \frac{\ln c}{(1-c\nu)(4n-3)}]$, we have $\mathbb{E}e^{\lambda z^2} \le e^{cn\nu(1+cn\nu)\lambda}$.*

*Proof.* Since $z$ is a binomial random variable, we can write $z = x_1 + \cdots + x_n$, where $x_1, ..., x_n$ are i.i.d. Bernoulli random variables with parameter $\nu$. Therefore applying Lemma B.3 completes the proof. $\qquad\square$

**Lemma B.8** (Multinomial variable). *Consider a multinomial random variable $(n_1, ..., n_K)$ with parameters $n$ and $(p_1, ..., p_K)$. For any $\delta > 0$:*

$$\Pr\left(\sum_{i=1}^{K} p_i^2 > \sum_{i=1}^{K}\left(\frac{n_i}{n}\right)^2 + 2\sqrt{\frac{2}{n}\ln\frac{K}{\delta}}\right) < \delta$$

*Proof.* Observe that

$$\sum_{i=1}^{K} p_i^2 - \sum_{i=1}^{K} \left(\frac{n_i}{n}\right)^2 = \sum_{i=1}^{K} \left[ p_i^2 - \left(\frac{n_i}{n}\right)^2 \right] \tag{73}$$

$$= \sum_{i=1}^{K} \left[ p_i + \frac{n_i}{n} \right] \left[ p_i - \frac{n_i}{n} \right] \tag{74}$$

$$= 2 \sum_{i=1}^{K} \left( 0.5 p_i + \frac{0.5 n_i}{n} \right) \left( p_i - \frac{n_i}{n} \right) \tag{75}$$

$$\leq 2 \max_{i \in [K]} \left( p_i - \frac{n_i}{n} \right) \tag{76}$$

where the last inequality can be derived by using the fact that $\sum_{i=1}^{K} \left( 0.5 p_i + \frac{0.5 n_i}{n} \right) \left( p_i - \frac{n_i}{n} \right)$ is a convex combination of the elements in $\{ p_i - \frac{n_i}{n} : i \in [K] \}$, because of $1 = \sum_{i=1}^{K} \left( 0.5 p_i + \frac{0.5 n_i}{n} \right)$. Furthermore, since $n_i$ is a binomial random variable with parameters $n$ and $p_i$, Lemma 5 in [23] shows that $\Pr \left( p_i - \frac{n_i}{n} > \sqrt{\frac{2 p_i}{n} \ln \frac{K}{\delta}} \right) < \delta$ for all $i$. This immediately implies $\Pr \left( p_i - \frac{n_i}{n} > \sqrt{\frac{2}{n} \ln \frac{K}{\delta}} \right) < \delta$. Combining this fact with (76), we obtain $\Pr \left( \sum_{i=1}^{K} p_i^2 - \sum_{i=1}^{K} \left(\frac{n_i}{n}\right)^2 > 2 \sqrt{\frac{2}{n} \ln \frac{K}{\delta}} \right) < \delta$, completing the proof. $\square$

## C  Experimental setup

More details about clustering the training images:

- We first preprocessed the images following Pytorch[2]: The images are resized to $resize\_size = [256]$ using interpolation=InterpolationMode.BILINEAR, followed by a central crop of $crop\_size = [224]$. Finally the values are first rescaled to $[0.0, 1.0]$. Those operations are required for Pytorch pretrained models.
- For each run, we randomly choose 200 points in $[0.0, 1.0]^{C \times H \times W}$ to be the centroids, since each preprocessed image belongs to $[0.0, 1.0]^{C \times H \times W}$. Those centroids are used to build the small areas $\mathcal{Z}_i$ in the partition. Each training image $x$ will be assigned to area $\mathcal{Z}_i$ if it is closest to the centroid of $\mathcal{Z}_i$ amongst all centroids, according to the Euclidean distance.

---

[2]https://pytorch.org/vision/0.20/models/generated/torchvision.models.vit_b_16.html

