# OpenReview forum: "Non-vacuous Bounds for the test error of Deep Learning without any change to the trained models"
_NeurIPS.cc/2025/Conference — Submitted to NeurIPS 2025_

### Official Review · Reviewer_TCVt · 2025-07-01

**Clarity:** 3
**Significance:** 2
**Originality:** 2
**Rating:** 4
**Confidence:** 3

**Summary:**

The generalization abilities of modern neural networks remain partially understood: their empirical performance is often not aligned with explanations provided by learning theory. This gap between theory and practice has motivated active research aimed at formally characterizing the generalization ability of deep learning models, typically by deriving population error bounds that are both computable and consistent with practical observations. This paper contributes to this line of research by proposing a novel theoretical bound on the population error of a trained neural network. Similar to many generalization bounds, their bound is expressed as the sum of the training error and a complexity term; here, the complexity term depends on a partition of the data space and captures the model's local behavior across these regions (Theorems 3.1 and 3.2). This bound differs from prior work in several ways (see Table 1): it relies on milder assumptions, e.g., it does not require model compressibility or a finite hypothesis class, and it yields non-vacuous bounds even for modern overparameterized architectures trained on large datasets, unlike previous results which have mostly been useful for small, stochastic, heavily compressed models (e.g., Lotfi et al., 2024). The authors also introduce a complementary bound taking into account data transformations, which provides further insight into the relationship between population error and other model properties, such as its sensitivity to data augmentation (Theorem 3.3). Their strategy is validated empirically on large deep learning models pretrained on ImageNet and is claimed to be the first to produce a non-vacuous theoretical generalization bound at this scale for realistic practical settings.

**Questions:**

I would appreciate it if the authors could address my comments above, and in particular the following:

**Q1.** Is is possible to further investigate how the introduced bounds behave under different data geometries or partitioning strategies?

**Q2.** Similarly, how do the bounds change when applying other data augmentation techniques?

**Q3.** Can the proposed bounds be compared to existing generalization bounds on simpler or smaller-scale models, maybe even on synthetic datasets?

**Ethical Concerns:**

["NO or VERY MINOR ethics concerns only"]

**Final Justification:**

After the rebuttal and discussion, I still have concerns about the lack of intuitive insights on how different partitioning strategies and data geometries affect the bound, as well as the limited comparisons with standard bounds like PAC-Bayes or robustness-based ones. That said, the paper makes a sound and interesting contribution to the NeurIPS community. The authors’ rebuttal was fair, with additional experiments and clarifications. While certain claims feel overstated (e.g., the tightness of the bounds) and the aforementioned aspects could be better motivated, the overall work is relevant enough for the conference. Considering these factors, I have updated my rating to borderline accept.

**Limitations:**

The potential negative societal impact is adequately addressed in the paper checklist. The limitations could be acknowledged more explicitly, maybe through a short dedicated paragraph toward the end.

**Paper Formatting Concerns:**

No major formatting issues.

**Quality:**

2

**Strengths And Weaknesses:**

**Strengths:**

- This paper is well written and addresses an important and timely problem that is relevant to the NeurIPS community: the development of reliable and practical tools to better understand the generalization behavior of deep learning models.
- The exposition and motivation for the work are clear, and the related work is adequately discussed.
- I particularly appreciate the discussion that follows the main theoretical results, as it helps clarify the intuition behind the proposed bounds and makes the contributions more accessible to both theoreticians and practitioners.
- The empirical validation on ImageNet is a strong contribution of the paper: the authors evaluate their bounds on 32 deep learning models from PyTorch, pretrained on the large-scale ImageNet dataset, and show that the bounds are non-vacuous (i.e., they are lower than 100% and correlate with the test error for most models).

**Weaknesses:**

While the bounds are quite discussed in the text and show promising results on ImageNet, I find their validation limited.

1) Some aspects of the empirical analysis remain unclear to me. For instance, the influence of $\alpha$ on $Unc(\Gamma)$ reported in Figure 2 (left) seems to contradict the explanation given in l.258. Similarly, the conclusions drawn in Section 4.2 regarding the data augmentation-based bound (l.316-319) appear to apply to the results in Table 2 as well, which raises questions about the practical utility of that bound in this context. Furthermore, I find the conclusions in l.324-328 are rather vague.

2) I would appreciate a more diverse set of experiments, potentially including simpler models or settings that may trade off complexity for greater interpretability. For example, since the proposed bounds rely on quantities related to how the data space is partitioned and structured, it would be relevant to include ablation studies that explore more this aspect. Empirical results illustrating how the bounds behave under different data geometries (aligned with the theoretical discussion in l.252-259) could significantly strengthen the paper. It would also be interesting to explore a wider range of data augmentation techniques beyond the injection of Gaussian noise. Currently, Tables 2 and 3 take up considerable space but seem to support similar conclusions; they could potentially be moved to the supplementary material in favor of introducing these new experiments that better illustrate the strengths and limitations of the proposed bounds.

3) The proposed bounds are not compared to any existing bounds in the empirical analysis. While I understand that evaluating prior bounds on large-scale, pretrained models may be difficult or even impossible, it seems feasible to include a comparison on smaller or stochastic neural networks. This would help further clarify the practical advantages/limitations of the contributions relative to prior work, even if only at a smaller scale or under other restrictive assumptions such as model compressibility or robustness (which, by the way, are concepts that seem to correlate with Theorem 3.3).

---

> ### Author Rebuttal · Authors · 2025-07-31
>
> We sincerely thank the reviewer for their thoughtful and constructive comments. We truly appreciate the time and effort spent on evaluating our work. The questions and suggestions raised have helped us clarify our contributions. Below, we respond to each point in detail and provide additional insights or empirical evidence.
>
> ### 1. Main questions
>
> **Q1:** *How the introduced bounds behave under different data geometries or partitioning strategies?*
>
> We thank the reviewer for this insightful question. Analyzing the effect of data geometry and partitioning strategies is indeed challenging, particularly in high-dimensional settings with unknown data distributions. To address this, we designed two controlled ablation studies using a synthetic mixture model, described in detail as follows:
>
> - **Mixture model (MM)**: each sample $(x,y)$ is generated by
>     - Randomly pick an index $z \sim cat(\theta)$, a categorical distribution with parameter $\theta =(1/K, ..., 1/K) \in R^K$
>     - Generate $x \sim Normal(\mu_z, \nu)$, a normal distribution with mean $\mu_z = (0, \pi*z) \in R^2$ and variance $\nu$
>     - Return class label $y= 1$ if $z$ is odd, and $y=0$ otherwise
> - The uncertainty term $Unc(\Gamma)=C\sqrt{\hat{u}\alpha \ln\gamma } + g_2$ is computed from a sample set $S$, with $K=100, \gamma=0.04^{-1/\alpha}$ and $\delta=0.01$ while varying $\alpha$. This term is the main part in Bound (3) that reflects the role of $\Gamma$.
>
> **Exploring different partitioning strategies:**
>
> We considered three types of partitions $\Gamma$:
>
> * **T1**: A uniform grid partition that divides the data space into equally sized regions. However, this strategy may not align with the actual data distribution, potentially resulting in regions with highly imbalanced probability measures.
> * **T2**: A partition formed by uniformly generating $K$ centroids to define the regions. Like T1, this method may not capture the underlying structure of the data.
> * **T3**: A partition where the centroids $\mu_1, ..., \mu_K$ of the mixture components are fixed as region centers. This approach tends to yield more balanced regions, where $P(\mathcal{Z}_i) \approx P(\mathcal{Z}_j)$ for all $i, j$, for small variances.
>
> To evaluate the quality of these partitions, we generated 100,000 i.i.d. samples $S$ from the MM (with $\nu = 1$) and computed $Unc(\Gamma)$ for varying $\alpha$. The results are as follows:
>
> | $\alpha$ | 4 |6 |8 |10 |
> | ---      | -- |-- |---|---|
> | T1 |1.0045 |0.8106 |0.7315 |0.6889 |
> | T2 |0.8723 |0.7302 |0.6722 |0.6409 |
> | T3 |**0.8450** |**0.7147** |**0.6615** |**0.6328** |
>
> Among the three strategies, T1 resulted in the highest uncertainty, while T3 consistently produced the lowest. These findings suggest that **partitions leading to balanced local measures (such as T3) are more favorable**, while those poorly aligned with the data distribution (T1, T2) lead to higher uncertainty. This empirical evidence supports our theoretical discussion on the importance of selecting meaningful partitions.
>
> **Exploring data geometries:**
> We further examine how the geometry of the data distribution influences the uncertainty term. To this end, we consider a mixture model with varying variances $\nu \in \\{10^0, 10^2, 10^4\\}$, while fixing the partition $\Gamma$ as T3, as described earlier. Notably, increasing $\nu$ to $10^4$ significantly alters the geometry of the mixture model compared to the case $\nu = 1$. The corresponding uncertainty values are reported below.
>
> | $\alpha$ | 4 |6 |8 |10 |
> | ---      | -- |-- |---|---|
> | $\nu=10^0$ |**0.8450** |**0.7147** |**0.6615** |**0.6328** |
> | $\nu=10^2$ |0.8458 |0.7151 |0.6618 |0.6331 |
> | $\nu=10^4$ |1.0142 |0.8393 |0.7680 |0.7295 |
>
> These results demonstrate that $Unc(\Gamma)$ can vary considerably depending on the geometry induced by the data distribution. When the partition $\Gamma$ does not align well with the data, the resulting local regions may have highly imbalanced probability measures. In such cases, the uncertainty can be large.
>
> **Q2:** *Other data augmentation techniques?*
>
> Thank you for the thoughtful question. To further investigate the impact of different data augmentation techniques, we conducted an additional set of experiments using **Brightness adjustment** as an alternative augmentation method. Specifically, we applied the `adjust_brightness` transform from Torchvision, varying the brightness factor between 1.4 and 2.5, and then computed Bound (4) accordingly. The results are presented below.
>
> | Model           | Train error | Test error | 1.4    | 1.6    | 1.8    | 2.0    | 2.5        |
> | --------------- | ----------- | ---------- | ------ | ------ | ------ | ------ | ---------- |
> | ResNet18 V1     | 0.2125      | 0.3024     | 0.6036 | 0.6559 | 0.7164 | 0.7560 | 0.8550     |
> | ResNet34 V1     | 0.1567      | 0.2669     | 0.4787 | 0.5323 | 0.5951 | 0.6402 | 0.7528     |
> | ResNet50 V1     | 0.1312      | 0.2387     | 0.4042 | 0.4568 | 0.5174 | 0.5830 | 0.7468     |
> | ResNet101 V1    | 0.1050      | 0.2263     | 0.3405 | 0.3923 | 0.4509 | 0.5130 | 0.6682     |
> | DenseNet121     | 0.1563      | 0.2557     | 0.4633 | 0.5064 | 0.5579 | 0.6284 | 0.8045     |
> | DenseNet169     | 0.1240      | 0.2440     | 0.3819 | 0.4238 | 0.4743 | 0.5450 | 0.7219     |
> | **DenseNet161** | *0.1048*    | **0.2286** | 0.3321 | 0.3734 | 0.4224 | 0.4907 | **0.6615** |
> | **DenseNet201** | *0.0980*    | **0.2310** | 0.3174 | 0.3626 | 0.4170 | 0.4893 | **0.6702** |
>
> We observed that Bound (4) exhibits behavior similar to that in the case of Gaussian noise: for most models, it shows a strong correlation with the test error. An exception occurs with **DenseNet161** and **DenseNet201**, where a relatively large brightness factor was needed to recover the superior performance of DenseNet161.
>
> Overall, these findings suggest that our bound responds consistently across different augmentation techniques.
>
> **Q3:** *Can the proposed bounds be compared to existing generalization bounds?*
>
> We sincerely thank the reviewer for this thoughtful suggestion. However, we would like to respectfully clarify that a direct numerical comparison between our bound and existing generalization bounds (e.g. PAC-Bayes or compression-based) can be inherently problematic and potentially misleading. As discussed in our response to Question Q7 from Reviewer 8XVE (see above), such comparisons may not be entirely fair due to fundamental differences in assumptions, applicability, and the way these bounds are computed.
>
> We also note that many existing bounds, aside from PAC-Bayes, typically require access to a validation set to approximate quantities that are not directly computable from the training data. In contrast, our bound is fully computable using the training data alone. This further complicates direct comparisons and may introduce additional biases.
>
> That said, we did carry out an additional comparison with robustness-based bounds proposed in [1, 2], which are also model-dependent. In this comparison, we applied our bound under the mild setting used in Table 2, while for [1, 2], we used $\delta = 0.05$ (corresponding to 95% confidence) and utilized the ImageNet validation set to approximate the intractable components. The results across 17 models are summarized below.
>
> | Model                     | Test error | Bound in [1] | Bound (8) in [2] | Our bound (3) |
> | ------------------------- | ---------- | ------------ | ---------------- | --------- |
> | ResNet18 V1               | 0.302      | 1.501        | 0.599            | 0.579     |
> | ResNet34 V1               | 0.267      | 1.437        | 0.553            | 0.523     |
> | ResNet50 V1               | 0.239      | 1.406        | 0.521            | 0.498     |
> | ResNet101 V1              | 0.226      | 1.377        | 0.504            | 0.472     |
> | ResNet152 V1              | 0.217      | 1.371        | 0.491            | 0.468     |
> | SwinTransformer B         | 0.164      | 1.323        | 0.432            | 0.431     |
> | SwinTransformer T         | 0.185      | 1.365        | 0.463            | 0.430     |
> | SwinTransformer B V2      | 0.159      | 1.322        | 0.421            | 0.466     |
> | SwinTransformer T V2      | 0.179      | 1.349        | 0.448            | 0.454     |
> | VGG13                     | 0.301      | 1.475        | 0.600            | 0.551     |
> | VGG13 BN                  | 0.284      | 1.478        | 0.580            | 0.559     |
> | VGG19                     | 0.276      | 1.444        | 0.565            | 0.528     |
> | VGG19 BN                  | 0.258      | 1.439        | 0.545            | 0.526     |
> | DenseNet121               | 0.256      | 1.432        | 0.527            | 0.523     |
> | DenseNet161               | 0.229      | 1.375        | 0.493            | 0.471     |
> | DenseNet169               | 0.244      | 1.398        | 0.513            | 0.490     |
> | DenseNet201               | 0.231      | 1.369        | 0.498            | 0.465     |
>
> Our findings show that our bound outperforms the existing robustness-based bounds in most cases, despite not relying on the validation set. This highlights the practical advantages and potential of our bound.
>
> [1] Robustness implies generalization via data-dependent generalization bounds. In ICML, 2022.
>
> [2] Gentle local robustness implies generalization. Machine Learning, 2025.
>
> ### 2. Other comments:
>
> **C1:** *the influence of $\alpha$ on $Unc(\Gamma)$ reported in Figure 2 (left) seems to contradict the explanation given in l.258*
>
> Line 258 mentions the certainty $1-\gamma^{-\alpha} -\delta$. Hence, there should be no contradiction.
>
> **C2:** *The conclusions drawn in Section 4.2 regarding the data augmentation-based bound (l.316-319) appear to apply to the results in Table 2 as well*
>
> Thank you for the observation. We agree that there are a few cases where a model with a smaller test error may have a larger bound compared to another model. These instances highlight some limitations of our bound.

---

> > ### Comment · Reviewer_TCVt · 2025-08-04
> >
> > Thank you for the detailed rebuttal.
> >
> > **Q1:** Thank you for providing this interesting toy example: it would be a nice addition to the paper, as it helps illustrate what the uncertainty term captures in relation to the partitioning strategy and data distribution. That said, the designed mixture model seems a bit unusual and far removed from typical deep learning settings (e.g., in terms of dimensionality), so it would be helpful to clarify why this particular setup was chosen.
> >
> > **Q2:**  Thank you for conducting this additional experiment. While this does not fully address how the bound behaves across a broader range of common data augmentation techniques, it offers another example where the bound seems to correlates well with the test error.
> >
> > **Q3:** : Regarding the comparison with PAC-Bayes bounds: I read your response to Question 7 from Reviewer 8XVE and have not fully understood it. One of your main arguments is that PAC-Bayes bounds are inherently for stochastic models and expressed in terms of expectations over the posterior distribution of parameters, whereas your bounds are for a fixed model $h$, so a direct comparison would not make sense. That said, isn't there a way to adapt the PAC-Bayes framework to make the comparison possible? For example by applying Markov’s inequality (to obtain a high-probability bound on a single model) or using a posterior distribution concentrated around the final model $h$.
> >
> > On robustness-based bounds: I appreciate the additional comparison; yet it remains unclear why your bound outperforms them in most cases.
> >
> > **C1:** Indeed, thank you for clarifying.
> >
> > **C2:** This is helpful to know and, in my opinion, should be stated explicitly in the paper to avoid potential confusion (such as the one I experienced during the review).

---

> ### Author Response · Authors · 2025-08-06
> **About the remaining concerns**
>
> Dear Reviewer TCVt
>
> We sincerely thank the reviewer for the thoughtful comments and valuable suggestions. Below, we address the remaining concerns.
>
> **Q1:**  *it would be helpful to clarify why this particular setup was chosen.*
>
> We used this mixture model as an intuitive example. The main reason is that one can choose some partitions which are easily verified to be bad or good, in terms of the balanced probability measures in local regions. For instance, partition T3 seems to be perfect. Furthermore, one can easily see the data geometry induced by such a mixture model.
>
> **Q3:** *Isn't there a way to adapt the PAC-Bayes framework to make the comparison possible? For example by applying Markov’s inequality (to obtain a high-probability bound on a single model) or using a posterior distribution concentrated around the final model $h$.*
>
> Thank you for your understanding. In fact, derandomizing PAC-Bayes bounds remains a challenging problem. While some prior work [3] has attempted this, such approaches typically impose additional assumptions, e.g., requiring the training algorithm to be deterministic. These assumptions do not align well with modern deep learning practices, which often involve inherently stochastic algorithms.
>
> Regarding the use of Markov's inequality: although it can be employed to convert an expectation $\mathbb{E}_g [F(P, g)]$ into a probability bound of the form $\Pr_g(F(P, g) \ge a) \le \frac{1}{a} \mathbb{E}_g[F(P, g)]$, the resulting probability is over the randomness of $g$. This differs fundamentally from the traditional PAC setting, where the probability is taken over the sampling of the dataset $S$, i.e., $\Pr_S(F(P, h) \ge a)$.
>
> Aside from recent work on LLMs [4,5], we are not aware of any method that allows for a direct and fair comparison between our bounds and PAC-Bayes bounds.
>
> **On robustness-based bounds:** We see two main limitations in prior work. First, the bound in [1] involves a robustness term defined as the worst-case difference between the losses of two samples within the same local region. As shown in [2], this term becomes vacuous for imperfect classifiers, a behavior we also observed empirically—e.g., for ImageNet models, the robustness term often equals 1. Second, the bound in [2] adopts a slightly looser uncertainty term compared to ours. Together, these factors explain why the robustness-based bounds tend to be weaker than ours, as reflected in our reported results.
>
> [1] Robustness implies generalization via data-dependent generalization bounds. In ICML, 2022.
>
> [2] Gentle local robustness implies generalization. Machine Learning, 2025.
>
> [3] A general framework for the practical disintegration of pac-bayesian bounds. Machine Learning, 2024.
>
> [4] Non-vacuous generalization bounds for large language models. In ICML, 2024.
>
> [5] Unlocking tokens as data points for generalization bounds on larger language models. In NeurIPS, 2024.
>
> **C2:** *This is helpful to know and, in my opinion, should be stated explicitly in the paper*
>
> Sure, we will add it in the revised version. We sincerely thank the reviewer for helpful suggestions.

---

> > ### Author Response · Authors · 2025-08-09
> >
> > Dear Reviewer TCVt
> >
> > As the discussion period will end shortly, we would like to know whether all of your concerns are fully addressed?
> >
> > Again, we appreciate your overall positive assessment and your suggestion to polish the claims and discussions. We will revise the paper accordingly to better reflect its limitations while emphasizing its practical relevance and theoretical novelty. We hope these clarifications and improvements will strengthen your confidence in the contribution and motivate a more favorable score.
> >
> > Best regards
> >
> > The authors

---

> > > ### Comment · Reviewer_TCVt · 2025-08-09
> > >
> > > Dear authors,
> > >
> > > Thank you for providing the additional clarifications and for considering my feedback. I have no further questions at this stage, and will continue the discussion with the other reviewers before finalizing my assessment.
> > >
> > > Best,

---

### Official Review · Reviewer_Nh1T · 2025-07-02

**Clarity:** 3
**Significance:** 3
**Originality:** 3
**Rating:** 4
**Confidence:** 3

**Summary:**

A pair of new tractable generalization bounds is proposed. These bounds are based on partitioning the data space into a union of disjoint sets; generally, tighter bounds are achieved when the partition results in a uniform distribution of samples across the sets. The second bound incorporates data augmentation and is claimed to be better suited for comparing different models. The authors claim that these bounds are the first non-vacuous results applicable to realistic deep neural networks; experimental results are provided to support this claim.

**Questions:**

1. How does the need to calculate $C$ from (1), (3) and (4) affects the applicability of your bounds in general case?
2. Have you tried other methods of acquiring $\Gamma$? How did it influence your results?

**Ethical Concerns:**

["NO or VERY MINOR ethics concerns only"]

**Final Justification:**

After the rebuttal, I decided to increase my score to "4: Borderline Accept." While I still believe that Weaknesses 1(a) (the bounds are too loose) and 1(b) (there are no experiments demonstrating that the proposed bounds capture overfitting or similar phenomena) remain unaddressed, my other concerns were alleviated.

In my opinion, describing such bounds as "non-vacuous" is still an overclaim. Nevertheless, the contribution is decent overall: the bounds are indeed non-vacuous in some cases and proved to be a slightly better predictor of test loss than training loss alone.

**Limitations:**

The checklist claims that limitations are discussed ("Yes"), but no dedicated section or justification is provided. Moreover, I see that justifications for other answers are missing too (I acknowledge that they are optional).

**Paper Formatting Concerns:**

Only a minor concern: the authors use vertical rules in tables, opposed to the strong suggestion from the style guide.

**Quality:**

2

**Strengths And Weaknesses:**

**Strengths:**

The paper tackles an important problem in learning theory. The background review is thorough and well-structured, and the theoretical derivations are rigorous. The findings are thoughtfully discussed. Additionally, the experimental scale is commendable, leveraging over 30 large vision models to validate the results.

**Weaknesses:**

1. Despite all the results, I still do not feel convinced about the non-vacuousness.
   - The gap between the test error and bound (3) remains substantial: even the optimized version never falls below the maximum observed test error ($\min(\text{Optimized}) = 36.482 > 30.242 = \max(\text{Test error})$). Moreover, bound (4) exceeds 100% in roughly one third of the cases reported in Table 3, raising doubts about its usefulness even for correlation analysis.
   - Only in some cases, the decrease in bound values appears linked to improved test accuracy rather than training accuracy, suggesting that the bounds may not be better predictors of test error than training error alone. Additional experiments could clarify whether the bounds truly capture overfitting or other scenarios where training error fails to predict test performance.
   - The presentation of the experimental results can be improved. While Tables 2 and 3 suggest a correlation between the bounds and test error, it remains unclear whether this correlation results from the explicit dependence on training error in (3) and (4). To improve clarity, I kindly ask the authors to explore more ways to present their findings: e.g., scatter plots with train/test errors being one axis (or marker color), and values of the bounds (or the difference between bounds and $F(S,h)$) being reported on the other axis.
2. The constant $C$ in (1), (3), and (4) requires taking a supremum over $Z$, which is non-trivial for many loss functions. While the authors avoid this issue by using the 0-1 loss, other choices of $\ell$ could make this computation challenging, limiting the bounds' applicability. This limitation should be explicitly discussed.
3. The selection of $\Gamma$ is crucial, as highlighted in the paper, yet the authors do not provide clear guidelines or analysis on how to choose it.

**Minor errors and typos:**

1. Abstract: "Our bounds uses"
2. Line 19: "Gimini"
3. Table 2: VGG13 seems to be in the wrong group.

---

> ### Author Rebuttal · Authors · 2025-07-31
>
> We sincerely thank the reviewer for their thoughtful and constructive comments. We truly appreciate the time and effort spent on evaluating our work. The questions and suggestions raised have helped us clarify our contributions and better position our results. Below, we respond to each point in detail and provide additional insights, empirical evidence, and clarifications where needed.
>
> ### 1. Main questions
>
> **Q1:** *How does the need to calculate C from (1), (3) and (4) affects the applicability of your bounds in general case?*
>
> Thank you for this thoughtful question. We would like to clarify that the constant $C$ in our bounds can be computed easily for many commonly used loss functions, particularly when the target function is bounded. This includes:
>
> - *Classification problems with $L$ classes:*
>   - For the 0-1 loss, $C = 1$
>   - For the absolute loss and squared loss, $C = 2$, assuming predictions and labels are represented by one-hot vectors
> - *Regression problems where outputs are bounded by a constant $A$:*
>   - For the absolute loss, $C = 2A$
>   - For the squared loss, $C = 4A^2$
>
> These cases cover many standard supervised learning settings in both theory and practice. For other loss functions such as the cross-entropy loss, the constant $C$ may be unbounded. In such situations, smoothing techniques (as explored in \[1, 2]) can be employed to make the bounds applicable.
>
> Overall, the ability to compute $C$ in a wide range of practical scenarios supports the broad applicability of our proposed bounds.
>
> [1] Non-vacuous generalization bounds for large language models. In ICML, 2024.
>
> [2] Unlocking tokens as data points for generalization bounds on larger language models. In NeurIPS, 2024.
>
> **Q2:** *Have you tried other methods of acquiring $\Gamma$? How did it influence your results?*
>
> We thank the reviewer for insightful question. Our experiments in Section 4 are really expensive. Therefore, we did not try to optimize our bounds accroding to partition $\Gamma$. An optimal choice of $\Gamma$ can produce tighter estimates about the true error.
>
> The result in the 6th column of Table 2 suggests that our bound is stable w.r.t. different choices of $\Gamma$, for a fixed granularity $K$.
>
> Inspired by the reviewer's question and to reveal more effects of $\Gamma$, we design a controlled ablation as follows:
> - Generate 100,000 i.i.d. samples  $\{(x_i,y_i)\}$ from the
> **Mixture model (MM)** as below:
>     - Pick randomly an index $z \sim cat(\theta)$, a categorical distribution with parameter $\theta =(1/K, ..., 1/K) \in R^K$
>     - Generate $x \sim Normal(\mu_z, 1)$, a normal distribution with mean $\mu_z = (0, \pi*z) \in R^2$ and variance $1$
>     - Return class label $y= 1$ if $z$ is odd, and $y=0$ otherwise
> - We considered three types of partitions $\Gamma$ with $K=100$:
>     - **T1**: A uniform grid partition that divides the data space into equally sized regions. However, this strategy may not align with the actual data distribution, potentially resulting in regions with highly imbalanced probability measures.
>     - **T2**: A partition formed by uniformly generating $K$ centroids to define the regions. Like T1, this method may not capture the underlying structure of the data.
>     - **T3**: A partition where the centroids $\mu_1, ..., \mu_K$ of the mixture components are fixed as region centers. This approach tends to yield more balanced regions, where $P(\mathcal{Z}_i) \approx P(\mathcal{Z}_j)$ for all $i, j$, for small variances.
> - We then compute the uncertainty term $Unc(\Gamma)=C\sqrt{\hat{u}\alpha \ln\gamma } + g_2$, with $\gamma=0.04^{-1/\alpha}$ and $\delta=0.01$. This term is the main part in our bound (3) that reflects the role of $\Gamma$. The results are below:
>
> | $\alpha$ | 10 |20 |30 |40 |50 |60 |70 |80 |90 |100 |
> | ---      | -- |-- |---|---|---|---|---|---|---|---|
> |T1 |0.6353 |0.5802 |0.5637 |0.5558 |0.5511 |0.5481 |0.5459 |0.5443 |0.5430 |0.5421 |
> |T2       |0.6409 |0.5855 |0.5689 |0.5610 |0.5563 |0.5532 |0.5510 |0.5494 |0.5481 |0.5471 |
> |T3       |0.6328 |0.5820 |0.5668 |0.5595 |0.5552 |0.5524 |0.5504 |0.5489 |0.5477 |0.5468 |
>
> For small $\alpha$, there is slight change in $Unc(\Gamma)$ for three different partitions. Furthermore, an increase in $\alpha$ can reduce the difference of the uncertainty. This suggests that our bounds can be more stable as $\alpha$ increases. This behavior is similar with that in Figure 2.
>
>
> ### 2. Other comments:
>
> **C1:** *The gap between the test error and bound (3) remains substantial. Moreover, bound (4) exceeds 100% in roughly one third of the cases reported in Table 3, raising doubts about its usefulness even for correlation analysis.*
>
> Thank you for this insightful comment. We fully agree with the reviewer that, despite certain strengths, Bound (3) remains loose in many cases. This highlights a clear opportunity for future improvements and refinement of our theoretical framework.
>
> That said, we would like to gently emphasize that *our bounds, while suboptimal, represent a meaningful step forward in bridging the long-standing gap between theoretical guarantees and practical performance in deep learning.* To the best of our knowledge, no existing theory can provide non-vacuous guarantees for the large-scale ImageNet models used in our experiments without significantly altering the models themselves.
>
> Regarding Bound (4), we would like to clarify that its primary goal is not to provide non-vacuous guarantees, but rather to support further exploration of model behavior and to enable meaningful comparisons between models. Although Bound (4) may appear imperfect, it still demonstrates a strong correlation with test error, as we illustrate below.
>
> | Bound | Correlation to Test error |
> | ----- | --- |
> | Bound (3) in Table 2 | 0.790 |
> | Bound (4) in Table 3 | 0.838 |
>
> A careful observation about Tables 2 and 3 suggests that within one architecture (e.g. ResNet) our bounds strongly correlate with the test error. Those observations suggest the practical utility of our bounds in practice.
>
> **C2:** *Only in some cases, the decrease in bound values appears linked to improved test accuracy rather than training accuracy, suggesting that the bounds may not be better predictors of test error than training error alone.*
>
> Thank you for raising this important point. We fully agree that our theoretical bounds do not always align perfectly with the test error observed in practice. Nonetheless, we would like to highlight that overall, the bounds exhibit a strong correlation with the test error, as demonstrated in the results below:
>
> | Bound | Correlation to Test error |
> | ----- | --- |
> | Bound (3) in Table 2 | 0.790 |
> | Bound (4) in Table 3 | 0.838 |
>
> A closer examination of Tables 2 and 3 suggests that *within a single architecture* (e.g., ResNet), our bounds correlate well with the test error. However, the correlation may weaken when comparing *across different architectures* (e.g., ResNet versus SwinTransformer).
>
> This observation further points to the need for improved architectural normalization in the bound formulation, which we view as a valuable direction for future work.
>
> **C3:** *The selection of $\Gamma$ is crucial, as highlighted in the paper, yet the authors do not provide clear guidelines or analysis on how to choose it.*
>
> Thank you very much for highlighting this important point. We agree that the choice of the partition $\Gamma$ is crucial to our bound.
>
> Our intention was to present a general framework that accommodates *a wide range of choices for $\Gamma$*, depending on the application or the domain knowledge available. In practice, we observe that simple partitions (e.g., uniform grid or clustering-based) can already yield meaningful and non-vacuous bounds, as demonstrated in our experiments. A better partition can lead to a tighter estimate of the test error.
>
> We appreciate the reviewer’s suggestion and agree that a more systematic study of partition selection would be a valuable direction for future work.
>
> **C4:** *The presentation of the experimental results can be improved.*
>
> Thank you for this helpful suggestion. This can strengthen our work and we'll revise it accordingly.

---

> > ### Comment · Reviewer_Nh1T · 2025-08-04
> >
> > Dear Authors,
> >
> > Thank you for your detailed response. Below, I provide my remaining comments.
> >
> > **Q1:** I appreciate the clarifications. However, it seems that even in the case of $A < +\infty$, the proposed bounds can quickly deteriorate as $A$ increases. Therefore, using such crude approximations for $C$ might be undesirable, and one might have to fall back to estimating $\sup_{z \in Z} \ell(h, z)$. That is why I want this problem to be treated as a limitation in the main text, with an according discussion on how to actually compute (or accurately estimate) $C$, and how it affects the practical applicability of the bound.
> >
> > **Q2:** Thank you for the additional experiments. It seems that the uncertainty term is decently stable w.r.t. $\Gamma$ in this particular low-dimensional synthetic setup. Is it possible to extend it to dimensionality of at least $100$ to make the setup more realistic? Is it possible to conduct a similar experiment with at least one real setup?
> >
> > **C1, C2:** Thank you for the clarifications. I still think that the paper is a bit overclaiming, given that the bounds are still too loose. My concerns are also shared by Reviewer 8XVE. In your response, you provided a correlation analysis for the bound vs. the test error. I kindly ask you to calculate the same correlations for the train error vs. test error and "bound minus train error" (i.e., the additional terms in (1), (3) and (4)) vs. test error. If the former is lower than the bound-test-error correlation, and the latter is at least positive, it might indicate that the bound captures something new. Otherwise, it might as well be that the bound behaves like "test error < train error + some big, almost constant term", hence the high correlation.
> >
> > In other words, can you please also compute the corresponding correlations for col. 5 vs col. 3 and col. 5 vs col. 6/7 minus col. 3 from Table 2 (and similar for Table 3)?
> >
> > **C3:** Understood, thank you.
> >
> > Currently, I think that the work is borderline. However, I kindly ask the authors to focus on my suggestions from the section **C1, C2**, as they are (a) doable within the time span of the rebuttal and (b) might present a stronger argument for the bounds.

---

> ### Author Response · Authors · 2025-08-06
> **Rebuttal by Authors**
>
> Dear Reviewer Nh1T
>
> Thank you very much for your insightful and helpful suggestions. We'd like to address the remaining concerns below.
>
> **Q1:** *using such crude approximations for $C$ might be undesirable, and one might have to fall back to estimating $\sup_{z \in Z} \ell(h, z)$. That is why I want this problem to be treated as a limitation*
>
> We fully agree with the reviewer that the examples we previously provided for $C$ are indeed crude approximations. In practice, incorporating domain knowledge or model-specific insights could lead to tighter, more meaningful estimates of $C$, potentially resulting in improved bounds. However, we also acknowledge that in some cases, computing or even approximating $C$ can be challenging. We therefore recognize this as a limitation of our current approach and appreciate the reviewer’s suggestion to highlight it as such.
>
> **Q2:**  *Is it possible to extend it to dimensionality of at least $100$ to make the setup more realistic? Is it possible to conduct a similar experiment with at least one real setup?*
>
> Thank you for this thoughtful question. To address it, we conducted an additional experiment in a 100-dimensional space:
>
> - We used the same mixture model as before, with the mean of the $z$-th Gaussian component set to $\mu_z = (0, 0, \ldots, 0, \pi \cdot z) \in \mathbb{R}^{100}$.
> - We considered the same partitions T2 and T3 as in the 2D setting, with $K = 100$. Partition T1 was omitted due to the difficulty of dividing the high-dimensional space into equal-sized regions.
>
> The results for $\text{Unc}(\Gamma)$ are summarized below.
>
> | $\alpha$ | 10 |20 |30 |40 |50 |60 |70 |80 |90 |100 |
> | ---      | -- |-- |---|---|---|---|---|---|---|---|
> | T2       |0.6452 |0.5883 |0.5713 |0.5632 |0.5584 |0.5552 |0.5530 |0.5513 |0.5500 |0.5490 |
> | T3       |0.6328 |0.5820 |0.5668 |0.5595 |0.5552 |0.5524 |0.5504 |0.5489 |0.5477 |0.5468 |
>
> We observe that increasing the dimensionality from 2 to 100 slightly affects the results for partition T2, while the results for T3 remain largely unchanged. This stability in T3 is likely due to its ability to better balance probability mass across local regions. Consistent with our earlier findings, we also see that increasing $\alpha$ reduces uncertainty differences, suggesting improved bound stability at higher $\alpha$ values.
>
> *Regarding real-world datasets:* since the underlying data distribution is unknown, constructing an effective partition similar to T3 is non-trivial without additional domain knowledge or assumptions. As such, replicating the exact experimental setup is challenging. However, we have computed our bound (3) under a range of more practical partition strategies:
>
> - Using randomly generated centroids,
> - Varying the granularity parameter $K$.
>
> Figure 2 presents results for the ImageNet dataset. Notably, for a fixed $K$, our bound (3) remains stable across different random partitions, further supporting the robustness of our method.
>
> **Q3:** *calculate the same correlations for the train error vs. test error and "bound minus train error" (i.e., the additional terms in (1), (3) and (4)) vs. test error.*
>
> Thank you for your insightful suggestion.
>
> We note that with our current choice of \$C = 1\$, the quantity *Bound (3) – train error* becomes a constant across models. As a result, computing its correlation with the test error is not meaningful. Instead, we focus on analyzing the correlations between *Bound (1) – train error* and *Bound (4) – train error* with the test error. The results are summarized below.
>
> | Quantity       |Correlation to test error |
> |----------------------|--------|
> | Train error | 0.7899 |
> | Bound (4) - Train error | 0.8525 |
> | Bound (1.T) - Train error | 0.7926 |
> | Bound (1.V) - Train error | 0.9918 |
>
> In particular, *Bound (1.T)* refers to the approximation of Bound (1) using the ImageNet training set, while *Bound (1.V)* uses the validation set.
>
> The results show that the uncertainty term in Bound (1) and the robustness term in Bound (4) capture meaningful characteristics of the trained models and correlate strongly with the test error. This highlights the practical relevance of our bounds for performance estimation.
>
> While Bound (3) offers a formal guarantee on test error, it does not exhibit the same level of correlation, and is therefore less effective in this specific empirical analysis.
>
> *We hope that those further information and discussions can fully address all the concerns of the reviewer. We will be happy to answer any further question.*

---

> > ### Comment · Reviewer_Nh1T · 2025-08-08
> >
> > Dear Authors,
> >
> > Thank you for additional experiments. I am now more inclined to increase my score. I will follow the discussion with other Reviewers to finalize my thoughts.

---

> ### Author Response · Authors · 2025-08-08
> **About the remaining concerns**
>
> Dear Reviewer Nh1T
>
> We would like to sincerely thank you for your positive feedback. We'll be happy to discuss further any more concern.
>
> **For the constant $C$:**
> We acknowledge that the constant $C$ in our bounds is a valid concern and appreciate the reviewer for raising this point. We recognize that identifying an appropriate value is not always straightforward and may affect the tightness of the bound. We will revise the manuscript to clearly state this limitation.
>
> We would also like to note that this challenge is not unique to our work, but is shared by many existing generalization bounds. For example, the bounds in [3,4] include a constant $\Delta$ that bounds the range of the loss function. Similarly, the bounds in [5,6] rely on a constant $C$ analogous to ours. The bound in [7] further requires knowledge of the Lipschitz constant of the loss, while [8] depends on the total Lipschitz stability constant $\gamma$ of the learning algorithm, both of which are typically harder to compute than our $C$.
>
> We hope this context helps clarify that while the constant presents a limitation, it is a common and often unavoidable aspect of generalization theory.
>
> Again, we appreciate your overall positive assessment and your suggestion to polish the claims and discussions. We will revise the paper accordingly to better reflect its limitations while emphasizing its practical relevance and theoretical novelty. We hope these clarifications and improvements will strengthen your confidence in the contribution and motivate a more favorable score.
>
> **Reference:**
>
> [3] Non-vacuous generalization bounds for large language models. In ICML, 2024.
>
> [4] Unlocking tokens as data points for generalization bounds on larger language models. In NeurIPS, 2024.
>
> [5] Robustness implies generalization via data-dependent generalization bounds. In ICML, 2022.
>
> [6] Gentle local robustness implies generalization. Machine Learning, 2025.
>
> [7] Slicing Mutual Information Generalization Bounds for Neural Networks. In ICML, 2024.
>
> [8] Algorithmic Stability Unleashed- Generalization Bounds with Unbounded Losses. In ICML, 2024.

---

### Official Review · Reviewer_Uu4e · 2025-07-03

**Clarity:** 3
**Significance:** 3
**Originality:** 3
**Rating:** 4
**Confidence:** 4

**Summary:**

This paper provides non-vacuous bounds for the test error of (rather big) deep learning models under the classification settings. Compared with existing meaningful bounds based on the PAC-Bayes and mutual information, it does not require stringent assumptions and extensive modifications (e.g., compression, quantization) to the trained model of interest. Experimental results on a large class of modern NNs, pretrained by Pytorch on the ImageNet dataset, verify that the bounds are non-vacuous.

**Questions:**

1. Why evaluate the only trained model? What about the one near the trained model? Can this framework consider a hypothesis space?
Please give more explanations.
2. Please give a more intuitive explanation of why including the partition $\Gamma$ can help to improve or induce the desired bounds.

**Ethical Concerns:**

["NO or VERY MINOR ethics concerns only"]

**Final Justification:**

The authors have addressed parts of my concerns, including the intuitive explanation of why including the partition can help to improve or induce the desired bounds, the limitation of Theorem 3.2, and the inconsistency between theory and experimental results. Especially, I do not agree with the statement about Theorem 3.3, which is a looser (vacuous) bound to get more information. Overall, I think the authors may overclaim the result and slightly decrease the score.

**Limitations:**

Please see the Weakness part.

**Minor comments:**

Besides, this paper considers the $0/1$ loss instead of the surrogate loss in experiments (or theory), which can be expressed more clearly.

**Quality:**

3

**Strengths And Weaknesses:**

# Strengths
1. Overall, this paper is well-written. The related work about the learning theory techniques and the research on non-vacuous bounds for NNs are easy to follow.
2. The main claims about the theory are well supported. The theoretical results are right, and I have checked the proofs in detail. Compared with the previous work on non-vacuous bounds based on the PAC-Bayes and mutual information, this work has no stringent assumptions and extensive modifications to the trained model, which is a good advantage.
3. Experimental results on a large class of modern NNs pretrained on the ImageNet dataset verify that the bounds are non-vacuous.

# Weaknesses
1. Theorem 3.1 shows that the generalization of a model is only dependent on its training error since Unc($\Gamma$) is independent of the models, which can not explain why two models share the same training error but different test errors. Besides, the theory results show that a smaller training error implies a smaller test error, which is not always consistent with the experimental results, as illustrated in Table 2.
2. Although Theorem 3.2 can provide an attempt to address the mentioned issue of Theorem 3.1, the results are not very satisfactory. On one hand, theoretically, the bound (4) in Theorem 3.2 is looser than the one (3) in Theorem 3.1, which is weird to use loose bounds to compare different models. On the other hand, experimentally, the bound (4) is often vacuous, as illustrated in Table 3.
3. The theory results depend largely on the trained model and the partition $\Gamma$. (1) The generalization bounds only evaluate one hypothesis (the trained model), which might not be fair to the norm-based bound, which considers a hypothesis space including the trained one. Please explain this point. (2) Please give a more intuitive explanation of why including the partition $\Gamma$ can help to improve or induce the desired bounds.

---

> ### Author Rebuttal · Authors · 2025-07-31
>
> We would like to thank the reviewer for praising our work and valuable feedback. We would like to discuss about the remaining concerns below.
>
> ### 1. Main questions
>
> **Q1:** *Why evaluate the only trained model? What about the one near the trained model? Can this framework consider a hypothesis space? Please give more explanations.*
>
> We thank the reviewer for this insightful question. In our work, we focus on estimating the true error $F(P, h)$ of a specific trained model (or hypothesis) $h$. The motivation is that $F(P, h)$ directly characterizes the generalization performance of $h$ on unseen data. Accurately estimating this quantity is crucial for providing theoretical guarantees, especially when $h$ is intended to be deployed in real-world, safety-critical applications. For this reason, we evaluate trained neural networks in our experiments to assess the practical utility and tightness of our proposed bounds.
>
> While our framework is designed for evaluating individual models, it can also be applied to any fixed hypothesis, such as one in the neighborhood of a trained model. Additionally, although our bounds can, in principle, be used to analyze finite hypothesis spaces by evaluating each member individually, they are not directly applicable to infinite hypothesis spaces in the same way as some other frameworks (e.g., PAC-Bayes). Extending our approach to broader hypothesis spaces is an interesting direction for future research.
>
> **Q2:** *Please give a more intuitive explanation of why including the partition can help to improve or induce the desired bounds.*
>
> We appreciate the reviewer’s question and are happy to provide a more intuitive explanation. Including the partition $\Gamma$ contributes to the strength of our bound for three main reasons:
>
> - **Capturing local data properties**: The partition $\Gamma$ allows us to examine the data distribution at a finer granularity, effectively dividing the data space into local regions. This enables us to estimate how frequently i.i.d. samples fall into different parts of the space. In Appendix B.3, we develop novel analyses involving binomial and multinomial random variables to formalize this intuition. These local sampling characteristics are then summarized by the quantities $u$ and $g$ in our bound (1). A more fine-grained partition yields a richer summary of the underlying data distribution.
> - **Analyzing model behavior locally**: The partition also helps us understand how the model $h$ behaves across different regions of the input space. Rather than evaluating $h$ globally, we use $\Gamma$ to capture its local performance, which is again reflected in the quantity $g$. This local perspective provides a more nuanced understanding of the model’s generalization behavior.
> - **Approximating the true error locally**: Finally, the partition allows us to construct provable approximations of the true error within each region. By leveraging properties of binomial random variables (Appendix B.3), we can control the approximation error at the local level. These local estimates are then aggregated to produce the overall bound in (1).
>
> In summary, the partition enables a more detailed exploration of both the data distribution and model behavior, which in turn leads to sharper and more informative generalization bounds.
>
> ### 2. Other comments:
>
> **C1:** *Theorem 3.1 shows that the generalization of a model is only dependent on its training error since $Unc(\Gamma)$ is independent of the models, which can not explain why two models share the same training error but different test errors.*
>
> The reviewer's concern may refer to Theorem 3.2. We totally agree with the reviewer about the limitation of bound (3) in Theorem 3.2. We also discuss this in our paper. The main reason comes from the use of $g_2$ to upper bound $g$ in Theorem 3.1 (see line 484). Removing this step can result in a better bound such as:
>
> $F(S,h)  + C\sqrt{\hat{u}\alpha \ln\gamma }  + g(\Gamma,h,\delta)$ &emsp;&emsp;&emsp;&emsp;&emsp; (5)
>
> where $g$ is a function of $h$. Therefore, this bound (5) can track the behavior of $h$ better than bound (3), suggesting a better way to compare two models. Unfortunately, $g$ cannot be computed exactly in practice, and approximation is needed.
>
> **C2:** *Besides, the theory results show that a smaller training error implies a smaller test error, which is not always consistent with the experimental results*
>
> Thank you for raising this important point. We fully agree that our theoretical bounds do not always align perfectly with the test error observed in practice. Nonetheless, we would like to highlight that overall, the bounds exhibit a strong correlation with the test error, as demonstrated in the results below:
>
> | Bound | Correlation to Test error |
> | ----- | --- |
> | Bound (3) in Table 2 | 0.790 |
> | Bound (4) in Table 3 | 0.838 |
>
> A closer examination of Tables 2 and 3 suggests that *within a single architecture* (e.g., ResNet), our bounds correlate well with the test error. However, the correlation may weaken when comparing *across different architectures* (e.g., ResNet versus SwinTransformer).
>
> This observation further points to the need for improved architectural normalization in the bound formulation, which we view as a valuable direction for future work.
>
> **C3:** *Although Theorem 3.2 can provide an attempt to address the mentioned issue of Theorem 3.1, the results are not very satisfactory. On one hand, theoretically, the bound (4) in Theorem 3.2 is looser than the one (3) in Theorem 3.1, which is weird to use loose bounds to compare different models. On the other hand, experimentally, the bound (4) is often vacuous, as illustrated in Table 3.*
>
> We sincerely appreciate the reviewer’s thoughtful feedback. We would like to clarify that the primary purpose of Bound (4) is **not to ensure non-vacuousness**, but rather to enable **meaningful comparison between models** and provide deeper insights into model behavior. While it may initially seem counterintuitive to use a looser bound for such comparisons, we have found this to be valuable in practice.
>
> In particular, Bound (4) exhibits a stronger empirical correlation with the test error than Bound (3), as shown before. This suggests that although Bound (4) may be looser in a theoretical sense, it can better reflect model performance in some practical settings, especially when the tightest theoretical bound does not align with observed generalization behavior.
>
> To further support this, we repeated the experiments in Table 3 under increased noise variance $\sigma$. The results (in %) are presented in the table below.
>
> | Model               | Train Error | Test Error | $\sigma=0.3$   | $\sigma=0.5$   | $\sigma=0.7$   | $\sigma=0.9$   | $\sigma=1.1$   |
> |---------------------|-------------|------------|--------|--------|--------|--------|--------|
> | RegNet Y 32GF linear| 10.56      | 15.38     | 90.76  | 143.18 | 169.04 | 179.58 | 190.12 |
> | *RegNet Y 32GF e2e*   | 7.13       | 13.16     | 70.82  | 128.15 | 164.53 | **180.19** | **195.86** |
> | *RegNet Y 32GF V2*    | 3.76       | 18.02     | 69.79  | 122.53 | 160.91 | **180.44** | **199.96** |
> | VIT B 16 linear     | 14.97      | 18.11     | 110.77 | 162.15 | 178.72 | 182.75 | 186.78 |
> | VIT L 16 linear     | 11.00      | 14.85     | 71.87  | 120.29 | 157.65 | 172.82 | 184.00 |
> | *VIT B 16 V1*         | 5.92       | 18.93     | 55.82  | **97.58**  | **135.16** | **161.55** | **187.94** |
> | *VIT L 16 V1*         | 3.47       | 20.34     | 52.21  | **97.67**  | **141.82** | **170.90** | **199.97** |
>
> We observed that, under appropriate noise settings, Bound (4) successfully recovers the relative quality of models, whereas training error and Bound (3) may provide misleading assessments.
>
> These findings indicate that **even a looser bound can be more informative in practice**, especially when used to understand or compare models under realistic conditions. We hope this perspective clarifies the intent and utility of Bound (4) in Theorem 3.3.
>
> **C4:** *The generalization bounds only evaluate one hypothesis (the trained model), which might not be fair to the norm-based bound, which considers a hypothesis space including the trained one.*
>
> We thank the reviewer for insightful comment. When suitably exploiting some prior knowledge about the hypothesis space, one can expect to obtain a tight estimate about the true error of the trained model $h$. However, none of the existing efforts can succeed at obtaining a non-vacuous norm-based bound.
>
>
> **C5:** *This paper considers the 0/1 loss instead of the surrogate loss in experiments (or theory), which can be expressed more clearly.*
>
> The theory part uses a bounded loss which is general. The experiment part uses the 0/1 loss to enable an easy link with error and accuracy, widely-used in practice.

---

> ### Comment · Reviewer_Uu4e · 2025-08-08
>
> Thank you for the detailed response.
>
> Q1&C4: I think this is the limitation of this work, which should be expressed more clearly. Please consider the model before the last (SGD) step of the end (corresponding to the trained model), which can also have promising performance in practice.
>
> Q2&C1&C2: Thanks for the clarification.
>
> C3: If the non-vacuous bound (3) is meaningful, I do not agree with the claim that a looser (vacuous) bound can be more informative in practice.
>
> C5: I have the same concern with the constant $C$ raised in Q1 of Reviewer Nh1T.
>
>
> Overall, despite the above concerns, I still think this paper provides a good attempt to address a very important problem in learning theory. I suggest that the authors polish the claims and discussions.

---

> ### Author Response · Authors · 2025-08-08
> **About the remaining concerns**
>
> Dear Reviewer Uu4e
>
> We sincerely thank the reviewer for the valuable suggestions. We appreciate your recognition of the importance of the problem addressed in this work and your acknowledgment of our contributions toward it. Below, we respond to your comments and clarify several points, with the hope that this may address your remaining concerns and support a higher evaluation.
>
> **Limitation and model before the final SGD step:**
>
> We agree that our model-dependent bounds can be limited in understanding a hypothesis space or learning algorithm. We will note this in the revised manuscript.
>
> While our bounds can be used to analyze some models in the course of training by some methods such as SGD, our focus on the *(finally-selected) trained model* aligns with practical deployment scenarios, where this is the model actually used in real-world systems. Estimating its true test error remains a crucial and unresolved challenge, and our work provides a step toward addressing it in a principled way.
>
> **C3 – Informative value of vacuous bounds:**
> We understand your disagreement and appreciate the chance to clarify. Our intention was not to suggest that vacuous bounds are preferable, but rather that certain looser bounds (e.g., Bound (4)) may correlate better with test error in practice due to how they encode relevant uncertainty information. Our experiments suggest that Bound (4) can have a better correlation than Bound (3). Such a phenomenon is in fact not new, since [1] found that norm-based bounds (which are highly vacuous [2]) may have a strong correlation with the test error. That said, we fully agree that non-vacuous bounds such as Bound (3) are ultimately more desirable and interpretable, and this is exactly the motivation for our work. We will revise the discussion to better reflect this nuance and avoid potential misinterpretation.
>
> **C5 – Constant in Bound (3):**
> We acknowledge that the constant in Bound (3) is a valid concern and appreciate the reviewer for raising this point. While $C$ can be easily computed in some settings, we recognize that identifying an appropriate value is not always straightforward and may affect the tightness of the bound. We will revise the manuscript to clearly state this limitation.
>
> We would also like to note that this challenge is not unique to our work, but is shared by many existing generalization bounds. For example, the bounds in [3,4] include a constant $\Delta$ that bounds the range of the loss function. Similarly, the bounds in [5,6] rely on a constant $C$ analogous to ours. The bound in [7] further requires knowledge of the Lipschitz constant of the loss, while [8] depends on the total Lipschitz stability constant $\gamma$ of the learning algorithm, both of which are typically harder to compute than our $C$.
>
> We hope this context helps clarify that while the constant presents a limitation, it is a common and often unavoidable aspect of generalization theory.
>
> *Again, we appreciate your overall positive assessment and your suggestion to polish the claims and discussions. We will revise the paper accordingly to better reflect its limitations while emphasizing its practical relevance and theoretical novelty. We hope these clarifications and improvements will strengthen your confidence in the contribution and motivate a more favorable score.*
>
> **Reference:**
>
> [1] Predicting trends in the quality of state-of-the-art neural networks without access to training or testing data. NATURE COMMUNICATIONS, 2021.
>
> [2] Stronger generalization bounds for deep nets via a compression approach. In ICML, 2018.
>
> [3] Non-vacuous generalization bounds for large language models. In ICML, 2024.
>
> [4] Unlocking tokens as data points for generalization bounds on larger language models. In NeurIPS, 2024.
>
> [5] Robustness implies generalization via data-dependent generalization bounds. In ICML, 2022.
>
> [6] Gentle local robustness implies generalization. Machine Learning, 2025.
>
> [7] Slicing Mutual Information Generalization Bounds for Neural Networks. In ICML, 2024.
>
> [8] Algorithmic Stability Unleashed- Generalization Bounds with Unbounded Losses. In ICML, 2024.

---

### Official Review · Reviewer_8XVE · 2025-07-03

**Clarity:** 2
**Significance:** 2
**Originality:** 3
**Rating:** 3
**Confidence:** 2

**Summary:**

The paper gives a new computable bound on the true test error of any trained neural network without modifying the model. It partitions the input space, links test error to training error plus two data driven terms, and then replaces the unknown quantities with counts from the training set. A variant with data augmentation measures robustness. The method needs only i.i.d. data and bounded loss. Applied to thirty two ImageNet models it yields nonvacuous guarantees that are at most about three times the actual test error, a first at this scale without compression or quantization.

**Questions:**

I am not a huge fan of the statement “it is largely unclear about why deep NNs generalize well unseen data”, please consider the follow on work to your citation [30] Non-Vacuous Generalization Bounds for Large Language Models recently published at ICLR. I would want a rebuttal to their argumentation (essentially that we can bound the error of quantised models and that quantisation error decreases with scale provably) in the text.

For Table 2, why do you have Acc@1 and test error where test error is just 100-acc@1? Please remove

As for Table 3, why do we need Gaussian noise on the training images? How do the computed bounds alter as a function fo the sigma chosen.

Bound is so large as to be useless, can we infer anything useful from the bound is there at least a correlation between bound size and test error? Can we plot this maybe in Figure 3c?

Lower priority questions:

Explain how choosing K and alpha with a grid search on the training set still allows the bound to hold at the nominal ninety five percent confidence. If possible show a union bound calculation or use a small tuning split to justify that the optimized numbers in Table 2 remain valid.
Describe in full how the input space partition is built. State the feature representation used for clustering, the distance metric, the clustering algorithm, the random seed strategy, and report any sensitivity the final bound has to these choices.
Provide a direct numerical comparison on a smaller network where a known PAC Bayes or compression bound can be computed so readers can see the quantitative improvement your method offers.
Clarify why the Gaussian noise robustness bound does not track test error for certain RegNet and ViT models and whether another transformation such as small rotations, blur, or adversarial perturbations would restore the correlation.
Suggest concrete analytical refinements for shrinking the gap between the reported bound and the true error, for example by using sharper concentration inequalities, a data specific value of C, or incorporating a notion of model margin.

**Ethical Concerns:**

["NO or VERY MINOR ethics concerns only"]

**Final Justification:**

Honestly the work seems to me to be very borderline, if I could give the work a 3.5 I would. I will not fight for the paper to be accepted or rejected, but I fall somewhat closer onto the reject category than accept. The Authors have clearly done a reasonable job of a rebuttal (without being allowed to address via an updated paper), but my main concerns remain and have co-authored works where they make comments on quantised models, their assessment on that factor is simply wrong (you can prove the difference between the two models drops with scale) and is not adequately addressed in their paper relative to their contribution.

**Limitations:**

yes

**Quality:**

3

**Strengths And Weaknesses:**

Strengths: The theory is sound, with clear assumptions (only i.i.d. data and bounded loss) and complete proofs that rely on well-known concentration tools. The new partition-based analysis and the data-driven “distribution complexity’’ term are original contributions, and the resulting bounds are computable from the training set alone. Experiments on thirty-two ImageNet networks confirm the bounds are non-vacuous even for models exceeding six hundred million parameters, something no prior work achieved without compressing the model. The writing is mostly clear, each theorem is followed by intuition, and the work closes an important gap between theory and practice, giving it high significance for both theoreticians and practitioners.
Weaknesses: The numerical bounds remain loose, often two to three times the true test error, limiting immediate practical value. Choosing the partition size K and confidence parameter α is done by grid search on the same data, so the advertised ninety-five percent confidence may not strictly hold after tuning. A direct empirical comparison with existing PAC-Bayes or compression bounds on a small model is missing, making the quantitative advance harder to gauge. The robustness variant works unevenly across architectures and offers little guidance on selecting a suitable data transformation. Minor clarity issues persist, such as typos and a brief description of the clustering procedure that defines the partition.

---

> ### Author Rebuttal · Authors · 2025-07-31
>
> We sincerely thank the reviewer for their positive evaluation and valuable suggestions. We truly appreciate the time and effort spent on evaluating our work. The questions and suggestions raised have helped us clarify our contributions. Below, we respond to each point in detail.
>
> ### 1. Main questions
>
> **Q1:** *please consider the follow on work to your citation [30] ... I would want a rebuttal to their argumentation*
>
> We'd like to thank you very much for reminding us on the writing. We'll rephrase it appropriately. Some of our arguments appear in lines 117--156.
>
> **Q2:** *For Table 2, why do you have Acc@1 and test error where test error is just 100-acc@1?*
>
> We included both columns to enable an easy verification with the source from Pytorch. Otherwise, using "test error" only may confuse the readers.
>
> **Q3:** *As for Table 3, why do we need Gaussian noise on the training images? How do the computed bounds alter as a function of the sigma chosen.*
>
> Bound (4) needs a transformation for the training set. We used Gaussian noises as a simple transformation.
>
> Figure 3 shows how the bound (for some models) can change when $\sigma$ changes.
>
> **Q4:** *Bound is so large as to be useless, can we infer anything useful from the bound is there at least a correlation between bound size and test error? Can we plot this maybe in Figure 3c?*
>
> We thank the reviewer for the insightful question.
>
> We would like to kindly clarify that Bound (3) demonstrates strong alignment with test error in many cases, as evidenced in Table 2, and exhibits a clear correlation between its value and test error, as shown below:
>
> | Bound | Correlation to Test error |
> | ----- | --- |
> | Bound (3) in Table 2 | 0.790 |
> | Bound (4) in Table 3 | 0.838 |
>
> Regarding Bound (4) (reported in Table 3), we acknowledge that it is not designed to provide a tight estimate of the test error. This bound is introduced to **highlight certain aspects of model behavior**, particularly under perturbations or rare conditions. Because it is evaluated on perturbed samples, it is naturally expected to be looser than Bound (3). This increase is consistent with the well-known challenges of robustness and adversarial vulnerability in neural networks.
>
> We agree that visualizing this correlation could be helpful and will be happy to include an additional plot to illustrate the relationship between bound values and test error.
>
> **Q5:** *How choosing K and $\alpha$ with a grid search on the training set still allows the bound to hold at the nominal ninety five percent confidence.*
>
> We'd like to remind that the confidence in Theorem 3.2 is $1-\gamma^{-\alpha} -\delta$. In our experiments, we chose $\gamma = 0.04^{-1/\alpha}$ and hence $\gamma^{-\alpha} = 0.04$ for any given $\alpha$. Therefore, with $\delta=0.01$, the confidence is $1-\gamma^{-\alpha} -\delta = 0.95$.
>
> **Q6:** *How the input space partition is built. State the feature representation used for clustering, the distance metric, the clustering algorithm, the random seed strategy, and report any sensitivity the final bound has to these choices.*
>
> We've already described the details in Appendix C, page 20. The sensitivity w.r.t. the partition can be observed from 6th column in Table 2. It also can be observed from Figures 1 and 2, where each standard deviation was computed from 5 different partitions.
>
> **Q7:** *Provide a direct numerical comparison on a smaller network where a known PAC Bayes or compression bound can be computed so readers can see the quantitative improvement your method offers.*
>
> We sincerely thank the reviewer for the thoughtful suggestion. However, we would like to respectfully clarify that providing a direct numerical comparison between our bound and existing PAC-Bayes or compression bounds is inherently problematic and may lead to biased conclusions. The reasons are as follows:
>
> - Our bound directly estimates the expected error $F(P, h)$ of a specific model $h$, and can be computed **exactly** from an i.i.d. training set with bounded loss, without requiring any modification to the model or additional assumptions. In contrast, under the same conditions, existing PAC-Bayes or compression bounds are generally **not computable in exact form**. While this setting may appear to favor our bound, we argue that it is representative of practical scenarios where one wants to assess the reliability of a trained model.
> - Many classical PAC-Bayes bounds [1, 2] aim to estimate $E_g[F(P, g)]$, which captures the expected error over a hypothesis distribution rather than a particular trained model. While useful for understanding properties of the learning algorithm or hypothesis class, this quantity is not guaranteed to reflect the actual error of the final trained model $h$. As such, a bound on $E_g[F(P, g)]$ does not necessarily imply an upper bound on $F(P, h)$, making direct comparison with our bound inherently unequal.
> - Existing derandomization techniques for PAC-Bayes bounds [3] often rely on restrictive conditions, such as requiring the training algorithm to be deterministic. These assumptions are not reflective of common practice in modern deep learning, which typically relies on stochastic algorithms.
> - Some recent PAC-Bayes bounds [4, 5] that do target $F(P, h)$ require significant model compression followed by fine-tuning. In these cases, the bound applies to the compressed model $h'$, which may differ substantially from the original model $h$. This makes it **incompatible** with our setting, where we estimate the true error of the actual model $h$ as trained. Applying our bound to a compressed model $h'$ would introduce bias favoring PAC-Bayes approaches and would not fairly represent the quality of the original model, since compression may significantly degrade performance [4].
>
> For these reasons, we chose not to include PAC-Bayes or compression bounds in our experimental comparisons.
>
> To directly address the reviewer's suggestion, we computed our bound for **MobileNet**. The results (in \%) for two variants trained by PyTorch are as follows:
>
> | Model | Mild Bound (3) | Optimized Bound (3) |
> | --    | --        | -- |
> |MobileNet_V2_v1 | 53.87 |	50.24 |
> |MobileNet_V3_Large_v1 | 43.75 |	40.12 |
>
> For the same MobileNet architecture, [2] reported the best PAC-Bayes bound as 96.5\%, which is notably looser than ours. Meanwhile, [1] achieved a bound of 40.9\% for a **compressed MobileViT**, relying heavily on compression and quantization techniques, along with a validation set to estimate a data-dependent prior.
>
> These observations highlight the advantages of our bound, especially in avoiding reliance on additional components like validation sets or compression, while still yielding tighter estimates.
>
> [1] Pac-bayes compression bounds so tight that they can explain generalization. In NeurIPS, 2022.
>
> [2] Non-vacuous generalization bounds at the imagenet scale: a pac-bayesian compression approach. ICLR, 2019.
>
> [3] A general framework for the practical disintegration of pac-bayesian bounds. Machine Learning, 2024.
>
> [4] Non-vacuous generalization bounds for large language models. In ICML, 2024.
>
> [5] Unlocking tokens as data points for generalization bounds on larger language models. In NeurIPS, 2024.
>
> **Q8:** *Why the Gaussian noise robustness bound does not track test error for certain RegNet and ViT models?*
>
> We'd like to thank the reviewer for insightful comment. Table (4) presents bound (4) which does not recover the better performance of RegNet and ViT in some cases. We hypothesized that the noises may not be significant enough.
>
> To verify our hypothesis, we increase the variance $\sigma$ of the Gaussian noises and then measured the bounds again. The results (in %) are as follows.
>
> | Model               | Train Error | Test Error | $\sigma=0.3$   | $\sigma=0.5$   | $\sigma=0.7$   | $\sigma=0.9$   | $\sigma=1.1$   |
> |---------------------|-------------|------------|--------|--------|--------|--------|--------|
> | *RegNet Y 32GF e2e*   | 7.13       | 13.16     | 70.82  | 128.15 | 164.53 | **180.19** | **195.86** |
> | *RegNet Y 32GF V2*    | 3.76       | 18.02     | 69.79  | 122.53 | 160.91 | **180.44** | **199.96** |
> | VIT B 16 linear     | 14.97      | 18.11     | 110.77 | 162.15 | 178.72 | 182.75 | 186.78 |
> | VIT L 16 linear     | 11.00      | 14.85     | 71.87  | 120.29 | 157.65 | 172.82 | 184.00 |
> | *VIT B 16 V1*         | 5.92       | 18.93     | 55.82  | **97.58**  | **135.16** | **161.55** | **187.94** |
> | *VIT L 16 V1*         | 3.47       | 20.34     | 52.21  | **97.67**  | **141.82** | **170.90** | **199.97** |
>
> We observe that *a suitable variance will recover the better performance of a model.* Nonetheless, each architecture may require a different $\sigma$. The reason for this behavior may come from the special property of the network architecture. A careful observation about Tables 2 and 3 suggests that within one architecture (e.g. ResNet) our bounds strongly correlate with the test error. However, our bounds may not perfectly correlate with the test error when comparing across architectures (e.g., ResNet and SwinTransformer).
>
> **Q9:** *Suggest concrete analytical refinements for shrinking the gap between the reported bound and the true error.*
>
> We've already discussed some ideas in lines 335--342.
>
> ### 2. Other comments:
>
> **C1:** *The numerical bounds remain loose*
>
> Thank you for this insightful comment. We fully agree with the reviewer that, despite certain strengths, Bound (3) remains loose in many cases. This highlights a clear opportunity for future improvements and refinement.
>
> We'd like to gently emphasize that *our bounds, while suboptimal, represent a meaningful step forward in bridging the long-standing gap between theoretical guarantees and practical performance in deep learning.* To the best of our knowledge, no existing theory can provide non-vacuous guarantees for the large-scale ImageNet models used in our experiments without significantly altering the models themselves.

---

> > ### Comment · Reviewer_8XVE · 2025-08-06
> > **Further questions**
> >
> > Q1. Please give a full rebuttal of the given argument, here in writing so I can evaluate it. It is not about writing, it is about why the follow on work (showing decreasing quantisation error as a function of scale, theoretically and empirically) does not adequately address the concerns this paper is trying to address.
> >
> > Q2. Remove the redundant information.
> >
> > Q3. Which other transformations have been tried?
> >
> > Q4 Thank-you for the clarification.

---

> > > ### Author Response · Authors · 2025-08-07
> > > **About the remaining concerns**
> > >
> > > Dear Reviewer 8XVE
> > >
> > > We would like to sincerely thank the reviewer for further clarification. Below, we address the remaining concerns.
> > >
> > > **Q1:** *Why the follow on work ... does not adequately address the concerns this paper is trying to address.*
> > >
> > > We sincerely thank the reviewer for the insightful question. While the follow-up work in [1,2] represents meaningful progress, it provides only limited insight and guarantees regarding the *original trained model* of interest. Below, we explain why it does not fully address the concerns raised in this paper:
> > >
> > > * **Quantization error reduction does not imply improved test error bounds.** Many prior methods in quantization and compression aim to preserve the *training* error of the original model. Under certain assumptions, these methods can ensure that the compressed model $h_c$ achieves similar performance on the training set, i.e., $F(S, h_c) \approx F(S, h)$. For example, [1] demonstrates a trade-off between model size and training error, showing that increasing the size of the compressed model can reduce quantization error $F(S, h_c) - F(S, h)$. However, this only pertains to training error—**it does not offer guarantees for the *test* error**. We are not aware of any theoretical result that provides a computable, non-vacuous bound on $\epsilon := |F(P, h_c) - F(P, h)|$. The expected compression error $\epsilon$ is generally unknown and cannot be computed from the training data alone. As a result, a reduction in quantization error does not necessarily imply a tighter or more reliable bound on the true test error of the uncompressed model $h$.
> > >
> > > * **Test error bounds for the compressed model do not translate to the original model.** Some existing works [1,2] provide non-vacuous generalization bounds for the compressed model $h_c$. However, these bounds do not directly apply to the uncompressed model $h$, since the relationship between their respective test errors $F(P, h)$ and $F(P, h_c)$ remains uncertain. Without a theoretical handle on the compression error $\epsilon$, it is unclear how close the compressed model's test performance is to that of the original model. Thus, such results fall short of providing a reliable guarantee for the test error of the trained model $h$.
> > >
> > > In contrast, our work directly targets the estimation of the true error $F(P, h)$ of a given trained model $h$. This focus is crucial because $F(P, h)$ reflects the generalization ability of the actual model that will be deployed. A meaningful and theoretically grounded estimate of this quantity is essential for building **trustworthy** real-world systems. To our knowledge, the bounds in [1,2] do not address this core concern.
> > >
> > >
> > > [1] Non-vacuous generalization bounds for large language models. In ICML, 2024.
> > >
> > > [2] Unlocking tokens as data points for generalization bounds on larger language models. In NeurIPS, 2024.
> > >
> > >
> > > **Q2.** *Remove the redundant information.*
> > >
> > > Sure, we will remove it from the camera-ready version. Thank you for this helpful suggestion.
> > >
> > > **Q3.** *Which other transformations have been tried?*
> > >
> > > Thank you for the thoughtful question. To further investigate the impact of different data augmentation techniques, we conducted an additional set of experiments using **Brightness adjustment** as an alternative augmentation method. Specifically, we applied the `adjust_brightness` transform from Torchvision, varying the brightness factor between 1.4 and 2.5, and then computed Bound (4) accordingly. The results are presented below.
> > >
> > > | Model           | Train error | Test error | 1.4    | 1.6    | 1.8    | 2.0    | 2.5        |
> > > | --------------- | ----------- | ---------- | ------ | ------ | ------ | ------ | ---------- |
> > > | ResNet18 V1     | 0.2125      | 0.3024     | 0.6036 | 0.6559 | 0.7164 | 0.7560 | 0.8550     |
> > > | ResNet34 V1     | 0.1567      | 0.2669     | 0.4787 | 0.5323 | 0.5951 | 0.6402 | 0.7528     |
> > > | ResNet50 V1     | 0.1312      | 0.2387     | 0.4042 | 0.4568 | 0.5174 | 0.5830 | 0.7468     |
> > > | ResNet101 V1    | 0.1050      | 0.2263     | 0.3405 | 0.3923 | 0.4509 | 0.5130 | 0.6682     |
> > > | DenseNet121     | 0.1563      | 0.2557     | 0.4633 | 0.5064 | 0.5579 | 0.6284 | 0.8045     |
> > > | DenseNet169     | 0.1240      | 0.2440     | 0.3819 | 0.4238 | 0.4743 | 0.5450 | 0.7219     |
> > > | **DenseNet161** | *0.1048*    | **0.2286** | 0.3321 | 0.3734 | 0.4224 | 0.4907 | **0.6615** |
> > > | **DenseNet201** | *0.0980*    | **0.2310** | 0.3174 | 0.3626 | 0.4170 | 0.4893 | **0.6702** |
> > >
> > > We observed that Bound (4) exhibits behavior similar to that in the case of Gaussian noise: for most models, it shows a strong correlation with the test error. An exception occurs with **DenseNet161** and **DenseNet201**, where a relatively large brightness factor was needed to recover the superior performance of DenseNet161.
> > >
> > > Overall, these findings suggest that our bound responds consistently across different augmentation techniques.

---

### Decision · Program_Chairs · 2025-09-17

**Decision:**

Reject

**Comment:**

This paper introduces novel generalization bounds which depend on the regularity of the sampling distribution rather than that of the model. The main generalization bounds are in Theorem 3.1 and its empirical version Theorem 3.2. The theorems rely on a **partition of input space**. For ease of discussion, the details below relate to Theorem 3.1, though it is to be acknowledged that Theorem 3.2 makes the bound fully computable empirically. The final upper bound mainly consist in:

(1) terms related to the test error (e.g. the components of $g$ which involve the $a_i$s, which are defined as the expected loss of the model on partition component $i$).

(2) boilerplate terms of the form $\tilde{O}(\frac{1}{n})$ where $n$ is the sample size.

(3) "complexity terms", which relate to the complexity of the sampling distribution rather than that of the model. This includes part of the  quantity $u$ which is taunted as the main complexity term, including a component of the form $\sum_{i} p_i^2$ (here $p_i$ is the probability associate with each partition component). This also includes the even more intuitive term $\frac{|T|}{n}$, which shows that the "optimistic" component of the bound roughly corresponds to a sample complexity equal to $|T|$ (which is the number of partition components).


The reviewers praised the **originality of the approach**, the extensive experiments covering 30 large vision models (reviewer Nh1T) and the well-structured literature review. On the other hand, concerns were raised regarding whether the bound is truly non-vacuous and whether the grid search over the partition components is valid (reviewer  8XVE). The relatively loose quality of the bound was also mentioned, together with concerns with the constant $C$. Importantly, reviewer TCVt commented on the somewhat unrealistic or idealized nature of the underlying assumption. The authors provided a thorough rebuttal to most questions, leading several reviewers to raise their scores. In particular, I agree with the authors regarding the validity of the grid search procedure and side with the authors on the fact that the bounds are very interesting in their own right independently of whether their correlation to the test error is strong. I also find it an important improvement that the authors have demonstrated the existence of a correlation between their bounds and the test error in their rebuttal to reviewer 8XVE. I also really liked the **toy synthetic data example** provided in the reply to reviewer TCVt, which **should be included in the revised version**.  Some reviewers also complained of the fact that the bound cannot be directly compared to existing Pac-Bayesian bounds due to the fact that the latter don't apply directly to a single predictor. I have to side with the reviewers here: some bounds can be evaluated for a single hypothesis, including bounds outside the PAC Bayes category (see for instance reference [18] in the main paper, or more up-to-date examples such as [NN1] and the references therein).

Overall, I find the **main idea** of the paper **very interesting**. I also agree that the related works section is relatively well structured, though I believe it could be more comprehensive: there are only a couple of references for each category of bounds, and one could argue that a proper discussion of data-dependency should also incorporate a discussion of the Neural Tangent Kernel literature [NTK1, NTK2]. In particular, it is worth noting (as I did in the discussion with the reviewers), even if the data-distribution is restricted to an extremely low dimensional subspace, this will not be enough to make the "distribution complexity" term tend to zero. This is a significant limitation to be discussed.

 I also feel that the dependence on the parameter $|T|$ should be better discussed, as this appears to be one of the main distributional complexity terms. In particular, I think a simple non PAC-Bayes  parameter-counting argument might lead to similar bounds: since the RHS already includes terms involving weighted sums of the $a_i$s, there is an analogy between the approach provided and simply considering the class of models which are constant on each partition, which is also associated with a function class capacity of $|T|$ through a simple parametrization. The next revision may benefit from a comparison to this approach. Lastly, it is also worth mentioning much earlier work which also showed generalization bounds based on the complexity of the input space with no assumption other than Lipschitzness on the chosen model. Whilst the bounds' decay w.r.t. sample size is much slower in that reference in general, this may not be the case if one imposes stricter Lipschitz constant constraints on the model (e.g. L1 instead of L2).

Thus, although the main ideas are **highly original** and the paper is *very close to the borderline*, it would strongly benefit from **one more round of revision**, including additional comparisons to other related approaches, interpretation of each term, as well as a more thorough numerical comparison with other generalization bounds.





**References**


[NTK1]  Sanjeev Arora, Simon S. Du, Wei Hu, Zhiyuan Li, Ruosong Wang. Fine-Grained Analysis of Optimization and Generalization for Overparameterized Two-Layer Neural Networks. ICML 2019

[NTK2] Arthur Jacot, Franck Gabriel, Clément Hongler. Neural Tangent Kernel: Convergence and Generalization in Neural Networks. NeurIPS 2018.

[NN1] Graf et al. On measuring excess capacity in neural networks.


[Metric] Distance-Based Classification with Lipschitz Functions. von Luxburg and Bousquet. JMLR 2004.